# Conditional Forecasts and Proper Scoring Rules for Reliable and Accurate Performative Predictions

**Philip Boeken**
University of Amsterdam

**Onno Zoeter**
Booking.com

**Joris M. Mooij**
University of Amsterdam

## Abstract

Performative predictions are forecasts which influence the outcomes they aim to predict, undermining the existence of correct forecasts and standard methods of elicitation and estimation. We show that conditioning forecasts on covariates that separate them from the outcome renders the target distribution forecast-invariant, guaranteeing well-posedness of the forecasting problem. However, even under this condition, classical proper scoring rules fail to elicit correct forecasts. We prove a general impossibility result and identify two solutions: (i) in decision-theoretic settings, elicitation of correct and incentive-compatible forecasts is possible if forecasts are separating; (ii) scoring with unbiased estimates of the divergence between the forecast and the induced distribution of the target variable yields correct forecasts. Applying these insights to parameter estimation, conditional forecasts and proper scoring rules enable performatively stable estimation of performatively correct parameters, resolving the issues raised by Perdomo et al. (2020). Our results expose fundamental limits of classical forecast evaluation and offer new tools for reliable and accurate forecasting in performative settings.

## 1 Introduction

In many real-world settings, predictions influence the outcomes they seek to forecast. A forecast of high traffic may cause drivers to change routes; a grim economic outlook may shift investor behaviour (MacKenzie et al., 2007). These *performative forecasts* break the classical assumption that the target distribution is fixed — instead, the forecasts are part of the causal system. This performativity complicates the task of correct prediction, and undermines the standard foundations of scoring rules and statistical estimation. Moreover, correctness of a forecast doesn't need to align with desirability of the outcome (van Amsterdam et al., 2025).

If forecasts affect the target variable, correct forecasts need not exist: there may be no distributional fixed point where the forecast aligns with the induced outcome, as is the case in *self-defeating prophecies*. However, we show that the forecasting problem becomes well-posed when the forecaster makes *conditional predictions* — in particular, predictions given observed covariates that separate the forecast from the target: the target distribution becomes *forecast-invariant*, and the existence of a correct forecast is recovered.

This paper formalizes the performative forecasting problem in a causal framework and studies conditions under which correct forecasting is possible. We prove that even under forecast-invariance, classical proper scoring rules generally fail to elicit correct forecasts. We provide two concrete solutions: (i) in a decision-theoretic setting, proper scoring can be recovered under structural conditions on the causal graph, producing *incentive compatibility* of the forecasters' score and the agents utility; (ii) we introduce a proper scoring approach based on unbiased estimates of the divergence between the forecast and the induced distribution of the target variable. We also analyze how correct forecasting parameters can be stably learned from data in performative environments, resolving the problems raised by Perdomo et al. (2020).

39th Conference on Neural Information Processing Systems (NeurIPS 2025).

## 1.1 Related work and detailed contributions

Our analysis of the existence of correct performative forecasts extends the work of Oesterheld et al. (2023) to conditional forecasts and formalises it in a causal modelling framework (Pearl, 2009) (Section 2). We provide necessary and sufficient conditions for the existence of correct conditional forecasts (Theorem 1). There is a vast literature on proper scoring rules for eliciting correct (non-performative) forecasts, see e.g. Brier (1950); McCarthy (1956); Grünwald and Dawid (2004); Gneiting and Raftery (2007). To our knowledge, the only existing work on scoring rules for performative forecasts is in specific decision-theoretic settings (Othman and Sandholm, 2010; Chen et al., 2011, 2014; Oesterheld and Conitzer, 2020; Hudson, 2025). In Section 3 we introduce a general theory of proper scoring rules for performative forecasts without restricting how the forecast affects the target variable, contrary to what is done in the aforementioned literature. We then prove an impossibility theorem (2) about the non-existence of proper scoring rules in general performative settings, complementing an impossibility result of Othman and Sandholm (2010) that scoring rules are generally not counterfactually strictly proper in decision-theoretic settings. In Section 4 we introduce the decision-theoretic setup considered by the earlier mentioned authors. Here, we separately consider the *properness* of a scoring rule and its *incentive compatibility* with an agent's utility (which have been merged into a single notion in previous works (Chen et al., 2011; Oesterheld and Conitzer, 2020)), clarifying the different notions, and providing a solution to the issue raised by van Amsterdam et al. (2025) that correct forecasts can be harmful. We then provide the additional insight that 'utility scores' (Oesterheld and Conitzer, 2020) can be made *strictly* proper by adding a suitably scaled strictly proper scoring rule to the utility score; see Theorem 4. In Section 5 we introduce another resolution of finding proper scoring rules in performative settings by scoring based on unbiased estimates of the divergence between the forecast and the observed outcomes; see Theorem 5 and its corollary. We give examples of unbiased estimators of certain strictly proper divergences, for binary and continuous outcome variables. This is compared with an inverse probability weighting method of Chen et al. (2011). Finally, in Section 6 we place this theory in the framework of Perdomo et al. (2020) for parameter estimation in performative settings: we show that estimating parameters for conditional forecasts with scoring rule-based estimators resolves the issues raised by Perdomo et al. (2020): the estimated parameters don't change over successive retraining ('performative stability') and they induce a correct forecast ('performative optimality') (Theorem 6).

## 2 Existence of correct forecasts through conditioning

Let $\mathcal{Y}$ be any measurable sample space of variable $Y$, and let $\mathcal{P}_{\mathcal{Y}}$ be a set of probability measures on $\mathcal{Y}$. We model the effect of forecast $F \in \mathcal{P}_{\mathcal{Y}}$ on outcome $Y$ with a causal model $M$ (e.g. a causal Bayesian network or a structural causal model (Pearl, 2009)[1]), inducing a causal mechanism $P_M(Y \mid \mathrm{do}(F))$.

Let $\mathcal{M}_{\mathrm{p}}$ be the set of such causal models $M$, so whose graph is a subgraph of Figure 1 when projected onto $F$ and $Y$, and which satisfy $\{P_M(Y) : M \in \mathcal{M}_{\mathrm{p}}\} \subseteq \mathcal{P}_{\mathcal{Y}}$.[2]

Figure 1

In a non-performative setting, if the forecaster believes $P^*$ to be the true distribution, then forecast $F$ is *correct* if $F = P^*$. This extends to the performative setting as follows.

**Definition 1.** Given a causal model $M \in \mathcal{M}_{\mathrm{p}}$, we say that the forecast $F$ is *correct for $M$* if $F = P_M(Y \mid \mathrm{do}(F))$.

One can interpret correct forecasts as fixed-points of the mapping $F \mapsto P_M(Y \mid \mathrm{do}(F))$, see also Grunberg and Modigliani (1954); Simon (1954) or Oesterheld et al. (2023). A *self-fulfilling prophecy* is a forecast which causes itself to be correct (Merton, 1949). A correct forecast of $Y$ need not exist. For example, for $P_0, P_1 \in \mathcal{P}_{\mathcal{Y}}$ with $P_0 \neq P_1$, let $M$ be such that

$$P_M(Y \mid \mathrm{do}(F)) = \begin{cases} P_0 & \text{if } F = P_1 \\ P_1 & \text{otherwise,} \end{cases}$$

then $M \in \mathcal{M}_{\mathrm{p}}$ has no correct forecasts. This phenomenon is well known:

---

[1]An introduction to causal models is given in Appendix A.

[2]For ease of exposition we don't let forecast $F$ depend on covariates $X$, nor let it be confounded with $Y$. In Appendix E, we show that the results from the main paper extend to the setting where $F$ depends on covariates $X$, which are allowed to be confounded with $Y$.

**Example 1.** Given a causal model $M \in \mathcal{M}_\mathrm{p}$, a *self-defeating prophecy* is a forecast which causes itself to be incorrect, or more formally, a forecast $F$ for which $F \neq P_M(Y \,|\, \mathrm{do}(F))$. An example would be a forecast of rising Covid-19 incidences, causing policy makers to implement social distancing or vaccination programs, effectively mitigating many Covid-19 incidences. When it is impossible for the forecaster (the 'prophet') to be correct, this problem is sometimes referred to as the *prophet's dilemma*.

Let $A$ be a variable with discrete sample space[3] $\mathcal{A}$ and let $\mathcal{P}_{\mathcal{A} \to \mathcal{Y}}$ be the space of conditional distributions $F(Y \,|\, A) : \mathcal{A} \to \mathcal{P}_\mathcal{Y}, a \mapsto F(Y \,|\, A = a)$. A *conditional forecast* of $Y|A$ is a set of forecasts

$$F(Y \,|\, A) = \{F(Y \,|\, A = a) : a \in \mathcal{A}\} \in \mathcal{P}_{\mathcal{A} \to \mathcal{Y}}.$$

If the conditional forecast $F(Y \,|\, A) \in \mathcal{P}_{\mathcal{A} \to \mathcal{Y}}$ affects the conditioning variable $A$ and outcome $Y$ then we have a causal mechanism as is graphically depicted in Figure 2a. Let $\mathcal{M}_\mathrm{pc} \subseteq \mathcal{M}_\mathrm{p}$ be the set of such causal models $M$, so whose graph is a subgraph of Figure 2a when projected onto $F$, $A$ and $Y$. If $A$ is a set of variables, let $\mathcal{M}_\mathrm{pc} \subseteq \mathcal{M}_\mathrm{p}$ be the set of causal models whose graph is a subgraph of Figure 2a after merging the variables in $A$ into a single multi-dimensional variable (see Appendix A for a formal definition) and projecting onto $F$, $A$ and $Y$.

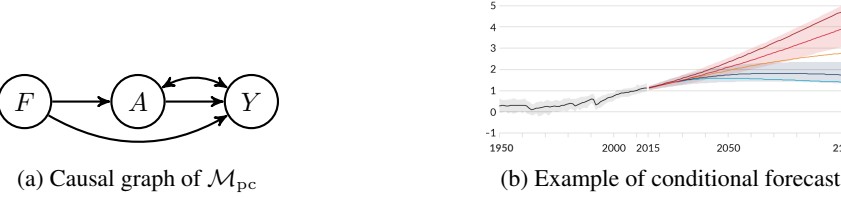

(a) Causal graph of $\mathcal{M}_\mathrm{pc}$        (b) Example of conditional forecast

Figure 2

**Example 2.** An example of a conditional forecast is the prediction of future average global temperatures given various scenarios of air-pollution, as provided by the IPCC (2021) (Figure 2b).

The notion of correctness of conditional forecasts is more nuanced than for marginal forecasts, since we can distinguish between correctness of the forecast for values of $A$ that can be observed, and the 'counterfactual' values of $A$ which cannot be observed.

**Definition 2.** Given a causal mechanism $M \in \mathcal{M}_\mathrm{pc}$, the conditional forecast $F(Y \,|\, A)$ is:

- *observationally correct* if $F(Y \,|\, A = a) = P_M(Y \,|\, A = a, \mathrm{do}(F))$ for all $a \in \mathcal{A}$ such that $P_M(A = a \,|\, \mathrm{do}(F)) > 0$;
- *counterfactually correct* if $F(Y \,|\, A = a) = P_M(Y \,|\, A = a, \mathrm{do}(F))$ for all $a \in \mathcal{A}$ such that $P_M(A = a \,|\, \mathrm{do}(F)) = 0$;
- *correct* if it is observationally and counterfactually correct.

Here, we pick the version of the conditional distribution $P_M(Y \,|\, A, \mathrm{do}(F)) = P_M(Y \,|\, \mathrm{do}(A, F))$ to ensure that $P_M(Y \,|\, A = a, \mathrm{do}(F))$ is well-defined for values $a \in \mathcal{A}$ with $P_M(A = a \,|\, \mathrm{do}(F)) = 0$. Note that the support of $A$, i.e. which values of $A$ can be observed after forecasting $F$, depends on the forecast $F$ itself. The distinction between observational and counterfactual correctness will be useful for defining various types of properness of scoring rules in Section 3, where one may require a forecast to only be observationally correct, or also counterfactually correct. In Appendix C we display a similar distinction for non-performative conditional forecasts.

A way to resolve the prophet's dilemma is to make a conditional forecast instead of a marginal forecast. If in the causal graph $G$ of model $M$ we have a d-separation $Y \perp^d_G F \,|\, A$, then the target distribution satisfies $P_M(Y \,|\, A, \mathrm{do}(F)) = P_M(Y \,|\, A)$, so it does not depend on the forecast itself. This for example holds if $A$ perfectly mediates $F$ and $Y$ and is not confounded with $Y$.[4]

---

[3]We assume that $\mathcal{A}$ is discrete to mitigate measure-theoretic problems.

[4]This has also been noted by Hudson (2025) for the special case where $A$ is an *action*: "*One method for avoiding performativity is to elicit predictions conditional on various actions that can be taken in response to the prediction. As this reaction is the causal pathway by which a prediction affects outcomes, conditioning on it removes the performative aspect.*"

**Definition 3.** We call the target $Y|A$ *forecast-invariant* in $\mathcal{M}' \subseteq \mathcal{M}_{\mathrm{pc}}$ if the target distribution does not depend on the forecast, so if $P_M(Y \mid A, \mathrm{do}(F)) = P_M(Y \mid A)$ for all $M \in \mathcal{M}'$.

**Example 3.** Consider the prophet's dilemma of predicting Covid-19 incidences (Example 1). Denote with $A = 0, 1$ whether policy makers implement social distancing, and let $Y = 0, 1$ respectively denote 'few' and 'many' subsequent Covid-19 incidences. If for a marginal forecast $F \in [0, 1]$ of the event $Y = 1$ we have the stylised mechanism $A = \mathbb{1}\{F > 1/2\}$ and $Y = 1 - A$, then $F \neq P(Y \mid \mathrm{do}(F))$ for all $F \in [0, 1]$, so the forecasting problem has no solution. However, if we consider a conditional forecast $F(Y \mid A = 0), F(Y \mid A = 1)$ with stylised mechanism $A = \arg\min_a F(Y \mid A = a)$ and $Y = 1 - A$, then the target $Y|A$ is forecast-invariant and the conditional forecast $F(Y \mid A = 0) = 1, F(Y \mid A = 1) = 0$ is correct, so the forecasting problem is well-posed.[5]

It is immediate that for a given target $Y|A$, forecast-invariance implies that $P_M(Y \mid A)$ is the unique correct forecast. Forecast-invariance is not necessary for the existence of an (observationally) correct forecast. One way of motivating forecast-invariance is by relying on the causal graph of the underlying system: the d-separation $Y \perp^d_G F \mid A$ implies forecast-invariance, as mentioned earlier. If no other assumptions are made, this d-separation even *characterises* forecast-invariance and whether the forecasting problem is well-posed, as shown in the following theorem. All proofs are provided in Appendix B.

**Theorem 1.** *Let $\mathcal{M}'_G \subseteq \mathcal{M}_{\mathrm{pc}}$ be the set of models compatible[6] with causal graph $G$, then the following are equivalent:*

*   *i) For every $M \in \mathcal{M}'_G$ there exists an observationally correct forecast of $Y|A$;*
*   *ii) For every $M \in \mathcal{M}'_G$ there exists a correct forecast of $Y|A$;*
*   *iii) $Y|A$ is forecast-invariant for all $M \in \mathcal{M}'_G$;*
*   *iv) $Y \perp^d_G F \mid A$.[7]*

## 3 Eliciting correct performative forecasts with proper scoring rules

Without appropriate incentives, forecasters may have motives to report an incorrect forecast. To address this, mechanisms must be designed to elicit correct forecasts: that is, to set the forecaster's incentives so that correct reporting is their optimal strategy. This mechanism is typically modelled as a *principal* eliciting a probabilistic forecast from a forecaster. They agree beforehand on a *scoring rule* $S : \mathcal{P}_\mathcal{Y} \times \mathcal{Y} \to \mathbb{R}$ which determines the payout of the forecaster: after each forecast $F$ and observation $y$, the forecaster receives $S(F, y)$ from the principal. A scoring rule is called proper if the forecaster maximizes the *expected score* $\overline{S}(F, P) := \int S(F, y) P(\mathrm{d}y)$ by reporting a correct forecast:

**Definition 4.** Scoring rule $S$ is *proper* relative to $\mathcal{P}_\mathcal{Y}$ if for all $F, P^* \in \mathcal{P}_\mathcal{Y}$ we have $\overline{S}(F, P^*) \leq \overline{S}(P^*, P^*)$, and *strictly proper* if the inequality is strict when $F$ is incorrect.

We sometimes refer to such a scoring rule as *(strictly) proper in the classical sense*. Well known examples of classical scoring rules are the *Bregman score* $S(F, y) := \varphi'(f(y)) + \int (\varphi(f(x)) - f(x)\varphi'(f(x))) \mathrm{d}x$ for $\varphi : \mathbb{R}^+ \to \mathbb{R}$ convex and differentiable and where $f$ is a density of $F$. The Bregman score is proper, and strictly proper if $\varphi$ is strictly convex. By picking $\varphi(f) = f \log f$, the Bregman score reduces to the *log-score* $S(F, y) = \ln(f(y))$. If $\mathcal{Y} = \{0, 1\}$, by picking $\varphi(f) = f^2$ and setting $f := F(Y = 1)$ the Bregman score reduces to the *Brier score* $S(F, y) = -(f - y)^2$ (Brier, 1950). If $\mathcal{Y} = \mathbb{R}^d$, another example is the *energy score* $S(F, y) = \mathbb{E}_{Y' \sim F} \|Y' - y\| - \frac{1}{2} \mathbb{E}_{Y, Y' \sim F} \|Y - Y'\|$, where $Y, Y'$ are independent random variables, and the expectations are to be computed analytically of via a sampling scheme (Székely and Rizzo, 2013). The energy score is strictly proper with respect to the class of distributions $P(Y)$ with $\mathbb{E}_P \|Y\|$ finite. For other examples we refer to Grünwald and Dawid (2004) and Gneiting and Raftery (2007).

---

[5]As an alternative solution to mitigate performativity in Covid-19 forecasting, Bracher et al. (2021) consider marginal, short-term forecasts with "*brief time horizons, at which the predicted quantities are expected to be largely unaffected by yet unknown changes in public health interventions.*"

[6]Meaning that the Markov property holds: $A \perp^d_G B \mid C \implies A \perp\!\!\!\perp_{P_M} B \mid C$ for all sets of variables $A, B, C$.

[7]We can have a d-connection $Y \not\perp^d_{G(M)} F \mid A$ but forecast invariance in $M$ due to faithfulness violations after intervention $\mathrm{do}(F)$ (Boeken et al., 2025). However, forecast invariance *for all $M \in \mathcal{M}'_G$* implies $Y \perp^d_G F \mid A$.

As straightforward extension of scoring rules to allow for conditional forecasts $F \in \mathcal{P}_{\mathcal{A} \to \mathcal{Y}}$, we refer to a map $S : \mathcal{P}_{\mathcal{A} \to \mathcal{Y}} \times \mathcal{A} \times \mathcal{Y} \to \mathbb{R}$ as a *conditional scoring rule*; after forecasting $F$ and observing $a$ and $y$, the forecaster receives $S(F, a, y)$ from the principal.[8] Any 'classical' scoring rule $S'$ induces a conditional scoring rule $S$ via $S(F, a, y) := S'(F(Y \mid A = a), y)$; if we consider a classical scoring rule for conditional forecasts, we always implicitly consider this induced conditional scoring rule. We define properness of conditional scoring rules for non-performative forecasts in Appendix C; this is generalised by Definition 5 in Section 3.1 below.

Most existing work on scoring rules assumes that the forecast does not affect the predicted variable, with an explicit mention in McCarthy (1956), assumption (iv): *"We assume that neither the forecaster nor the [principal] can influence the predicted event [...]"*. Our main interest lies in dropping this assumption, as is often required in practice (Perdomo et al., 2020; Boeken et al., 2024).

## 3.1 Defining properness of scoring rules for performative forecasts

To extend the theory of proper scoring rules to the performative setting we adhere to the same rationale as in the classical setting: if the forecaster knows that $M$ is the true model, then the expected score should be maximised at a correct forecast, where now the expected score of a forecast $F$ will be considered with respect to $P_M(Y \mid \mathrm{do}(F))$. Given a scoring rule $S : \mathcal{P}_{\mathcal{Y}} \times \mathcal{Y} \to \mathbb{R}$, a marginal forecast $F \in \mathcal{P}_{\mathcal{Y}}$ and causal model $M \in \mathcal{M}_{\mathrm{p}}$ the expected score is

$$\overline{S}_{\mathrm{p}}(F, M) := \int S(F, y) P_M(\mathrm{d}y \mid \mathrm{do}(F)).$$

For a conditional scoring rule $S : \mathcal{P}_{\mathcal{A} \to \mathcal{Y}} \times \mathcal{A} \times \mathcal{Y} \to \mathbb{R}$, conditional forecast $F(Y \mid A) \in \mathcal{P}_{\mathcal{A} \to \mathcal{Y}}$ and model $M \in \mathcal{M}_{\mathrm{pc}}$ the expected score is

$$\overline{S}_{\mathrm{pc}}(F, M) := \int S(F, a, y) P_M(\mathrm{d}a, \mathrm{d}y \mid \mathrm{do}(F)).$$

Recall Definitions 1 and 2 of the various types of correctness of performative (conditional) forecasts.

**Definition 5.** For marginal forecasts, a scoring rule $S$ is *proper* relative to $\mathcal{M}' \subseteq \mathcal{M}_{\mathrm{p}}$ if $\overline{S}_{\mathrm{p}}$ is maximised at a correct forecast, and *strictly proper* if all maximisers are correct. We call a conditional scoring rule

- *observationally (strictly) proper* relative to $\mathcal{M}' \subseteq \mathcal{M}_{\mathrm{pc}}$ if $\overline{S}_{\mathrm{pc}}$ is (only) maximised at observationally correct forecasts for all $M \in \mathcal{M}'$;
- *counterfactually (strictly) proper* relative to $\mathcal{M}' \subseteq \mathcal{M}_{\mathrm{pc}}$ if $\overline{S}_{\mathrm{pc}}$ is (only) maximised at counterfactually correct forecasts for all $M \in \mathcal{M}'$;
- *(strictly) proper* relative to $\mathcal{M}' \subseteq \mathcal{M}_{\mathrm{pc}}$ if it is observationally and counterfactually (strictly) proper, that is, if $\overline{S}_{\mathrm{pc}}$ is (only) maximised at correct forecasts for all $M \in \mathcal{M}'$.[9]

If we consider a class of models $\mathcal{M}'$ where $F$ does not affect $Y$, then Definition 5 is equivalent to Definition 4 and its conditional version (Appendix C). For a scoring rule to be (observationally) proper with respect to $\mathcal{M}'$, it is required that for every $M \in \mathcal{M}'$ there *exists* an (observationally) correct forecast $P^*$, signifying the subtleties of Section 2: **if one only makes assumptions about the causal graph, Theorem 1 implies that the target variable must be forecast-invariant for a scoring rule to be proper.** Note that if a correct forecast exists, any constant scoring rule is proper; throughout we only consider non-constant scoring rules.

## 3.2 Impossibility of eliciting correct performative forecasts with classical scoring rules

Few examples of conditional scoring rules exist in the literature; the most straightforward examples are conditional scoring rules $S$ induced by classical scoring rules $S'$ through $S(F, a, y) = S'(F(Y \mid A =$

---

[8]To our knowledge, the only related literature on scoring rules for conditional forecasts $F \in \mathcal{P}_{\mathcal{A} \to \mathcal{Y}}$ are Kadane et al. (1980); Schervish et al. (2009) in the non-performative setting, who consider conditional scoring rules of the form $S(F(Y \mid A = a), y)$, and Othman and Sandholm (2010); Chen et al. (2014); Oesterheld and Conitzer (2020) who consider scoring rules for conditional forecasts in the performative setting of the form $S(F, y)$, $S(F, P_M(A), a, y)$ and $S(F(Y \mid A = a), y)$ respectively.

[9]In a decision-theoretic context, Othman and Sandholm (2010) and Chen and Kash (2011) refer to scoring rules which are both observationally strictly proper and counterfactually proper as 'quasi-strictly proper' scoring rules.

$a), y)$. In this section we investigate whether one can expect these induced conditional scoring rules to be proper in the performative sense, even if they are strictly proper in the classical sense.

As remarked in the previous section, forecast-invariance is necessary for a scoring rule to be proper; however, it is not sufficient. Typically, the easier the forecasting problem, the higher the expected score of a correct forecast. If traffic jams are hard to predict on a highway called 'Route A', it can be beneficial for the forecaster to incorrectly predict a terrible traffic jam on Route A so that all cars will take an alternative, easier-to-predict Route B; the forecaster will benefit from a correct forecast on Route B, and will never be punished for the incorrect forecast on Route A. **In this mechanism the scoring rule is observationally strictly proper but not counterfactually proper**. The following example is adapted from Othman and Sandholm (2010):

**Example 4.** Let $\mathcal{A} = \mathcal{Y} = \{0, 1\}$ and $M$ be such that $Y \,|\, A$ is forecast-invariant, with $P_M(Y = 1 \,|\, A = 0) = 0.5, P_M(Y = 1 \,|\, A = 1) = 0.25$ and the 'decision rule' $A = \arg\max_a F(Y = 1 \,|\, A = a)$. If the forecaster reports the correct forecast $A = 0$ is chosen and the expected Brier score is $-0.5 \cdot (0.5)^2 - 0.5 \cdot (0.5)^2 = -0.25$, and if the forecaster reports $F(Y = 1 \,|\, A = 0) = 0.2, F(Y = 1 \,|\, A = 1) = 0.25$ then $A = 1$ is chosen the expected score is $-0.25 \cdot (0.75)^2 - 0.75 \cdot (0.25)^2 \approx -0.19$. The scoring rule is observationally strictly proper since for the observed value $A = 1$ the score is maximised at a correct forecast, but it is not counterfactually proper since it pays off to misreport for the value $A = 0$, which is not observed.

Inspired by the self-defeating prophecy, we can construct even more pathological examples where the scoring rule is neither observationally nor counterfactually proper. Consider a mechanism $P_M(A \,|\, \mathrm{do}(F))$ which can pick between two values of $A$: one for which $Y$ is easier to predict, giving a higher expected score, and one for which $Y$ is harder to predict, giving a lower expected score. If for any (close to) correct forecast the mechanism $P_M(A \,|\, \mathrm{do}(F))$ picks the harder-to-predict action, it is beneficial for the forecaster to misreport since this induces an easier-to-predict action, yielding a higher expected score. Hence, **in this mechanism the scoring rule is observationally and counterfactually improper**. Notably, the mechanism $P_M(A \,|\, \mathrm{do}(F))$ can be, but need not be, deterministic. In Appendix F.1 we provide examples and 3D-plots of this phenomenon.

The following theorem shows formally that in general performative settings, non-constant proper scoring rules don't exist. The proof relies on the phenomenon of the self-defeating prophecy, as just explained.

**Theorem 2.** *Let $\mathcal{M}' \subseteq \mathcal{M}_{\mathrm{pc}}$ be the set of models such that $Y \,|\, A$ is forecast-invariant, where $P_M(A \,|\, \mathrm{do}(F))$ has full support for all $F \in \mathcal{P_Y}$ and all $M \in \mathcal{M}'$, or where $P_M(A \,|\, \mathrm{do}(F))$ is deterministic for all $M \in \mathcal{M}'$. If $S$ is a scoring rule with respect to $\mathcal{P_Y}$ such that there are $P_0, P_1 \in \mathcal{P_Y}$ and forecast $\tilde{F} \in \mathcal{P_Y}$ with $\tilde{F} \neq P_1$ and $\overline{S}(P_0, P_0) < \overline{S}(\tilde{F}, P_1) < \overline{S}(P_1, P_1)$, then $S$ is not observationally proper for predicting $Y \,|\, A$ in $\mathcal{M}'$.*

Notably, this scoring rule is also not observationally proper for any larger class of models, for example when one does not require $Y \,|\, A$ to be forecast-invariant or if no assumptions are made on the mechanism $P_M(A \,|\, \mathrm{do}(F))$. We will cover two solutions which go beyond classical scoring rules: in Section 4 we consider a decision-theoretic class of models with a particular deterministic mechanism $P_M(A \,|\, \mathrm{do}(F))$ (the same for all $M$) for which proper conditional scoring rules exist, and in Section 5 we relax the setting by allowing the principal to score forecasts using estimated properties of the distribution $P_M(A, Y \,|\, \mathrm{do}(F))$. Another notable solution is given by Hudson (2025), who relaxes the setting by considering multiple forecasters who compete in a zero-sum game.

## 4 Properness and incentive compatibility in a decision-theoretic setting

Often, the forecast is provided to an agent who acts based on the forecasted information.[10] The quality and reliability of the forecast thus have a direct impact on the agent's behaviour and, consequently, on outcomes which may be of interest to the principal. More specifically, given a conditional forecast $F$, let the agent take an action which maximizes the expectation of a utility $U(a, y)$, so we have $A = a_F := \arg\max_{a \in \mathcal{A}} \int U(a, y) F(\mathrm{d}y \,|\, a)$.[11] This requires the forecast $F(Y \,|\, A)$ to be

---

[10] Note that this agent might be the principal herself.

[11] The action $a_F$ is often referred to as the *Bayes act*; see also Dawid (2007); Brehmer and Gneiting (2020); Dawid (2021).

a forecast of the *causal effect* $P_M(Y \mid \mathrm{do}(A))$, which holds in our setting since we assume that there is no confounding between $A$ and $Y$: let $\mathcal{M}_{\mathrm{dt}} \subseteq \mathcal{M}_{\mathrm{pc}}$ be the set of causal models $M$ with $P_M(A \mid \mathrm{do}(F)) = \delta_{a_F}$.

**Definition 6.** Given a utility $U$ and a set of causal models $\mathcal{M}' \subseteq \mathcal{M}_{\mathrm{pc}}$, we call a conditional scoring rule $S$ *incentive compatible*[12] with $U$ if for all $M \in \mathcal{M}'$ we have $\arg\max_F \overline{S}_{\mathrm{pc}}(F, M) = \arg\max_F \int U(a, y) P_M(\mathrm{d}a, \mathrm{d}y \mid \mathrm{do}(F))$.

The goal of the agent is to maximise its expected utility, and the goal of the forecaster is to maximise its expected score. As also noted by Othman and Sandholm (2010) and by van Amsterdam et al. (2025) for marginal forecasts, these two incentives don't necessarily align:

**Example 5.** Consider Example 4, interpreted as a forecast being given to an agent with utility $U(a, y) = y$; then indeed $A = a_F = \arg\max_a F(Y = 1 \mid A = a)$. The optimal action for the agent is $a^* = 0$, but to maximise the score the forecaster reports $F(Y = 1 \mid A = 0) = 0.2$ and $F(Y = 1 \mid A = 1) = 0.25$, forcing the agent to take the suboptimal action $a_F = 1$. Despite the forecast being observationally correct, it is not incentive-compatible with $U$.

It is not hard to see that every strictly proper conditional scoring rule is incentive compatible with any utility. Generalising the result of Othman and Sandholm (2010) (Theorem 4) to our setting, we moreover have that in $\mathcal{M}_{\mathrm{dt}}$, proper conditional scoring rules must have the form $S(F, a, y) = S(F(Y \mid A = a_F), a_F, y)$, so they may only depend on forecasts for unobserved actions through $a_F$, and they can never be counterfactually strictly proper.

Nevertheless, proper, incentive compatible conditional scoring rules exist; we will later show that they can even be made observationally strictly proper and incentive compatible. In the non-performative setting where marginal forecasts $F(Y)$ affects $A$ but where $A$ does not affect $Y$, it is well known that the *utility score* $S(F, y) := U(a_F, y)$ is proper (Dawid, 2007; Brehmer and Gneiting, 2020). In the performative setting with conditional forecast, we can straightforwardly consider the conditional scoring rule $S(F, a, y) = U(a_F, y)$; since the expected score then equals the expected utility, it is automatically incentive-compatible – not only in $\mathcal{M}_{\mathrm{dt}}$, but even for all $M \in \mathcal{M}_{\mathrm{pc}}$, so for any (possibly stochastic) policy $P_M(A \mid \mathrm{do}(F))$. As shown by Oesterheld and Conitzer (2020), this utility score is proper in $\mathcal{M}_{\mathrm{dt}}$:

**Theorem 3.** *The utility score is incentive-compatible with $U$, and proper if $Y|A$ is forecast-invariant in $\mathcal{M}' \subseteq \mathcal{M}_{\mathrm{dt}}$. If for a given causal graph $G$ one considers the class of models $\mathcal{M}'_G \subseteq \mathcal{M}_{\mathrm{dt}}$ which are consistent with $G$, then forecast invariance of $Y|A$ is* necessary *for the utility score to be proper.*

A main restriction of the utility score is that it is not observationally *strictly* proper: there can be multiple observationally incorrect forecasts which maximise the score. However, in the following we show that there does exist a conditional scoring rule which is incentive-compatible and observationally strictly proper. If the principal knows that the expected utility between any two actions differs with at least $\Delta > 0$, then the principal can elicit incentive-compatible and observationally correct forecasts by first scoring using the utility score to ensure that the optimal action $a^*$ is taken, and subsequently add a scaled strictly proper scoring rule $S'$ for $F(Y \mid A = a^*)$ to ensure that the maximum is uniquely attained at the observationally correct forecast $P_M(Y \mid A = a^*)$, where the magnitude of $S'$ is bounded by $\Delta$ to ensure that it does not incentivise a forecast which induces an action different than $a^*$. This utility gap $\Delta$ can for example be assumed if $A$ is a binary indicator of a 'high-risk, high-reward' action, like a risky operation or a bold marketing campaign.

**Theorem 4.** *Let $Y|A$ be forecast-invariant in $\mathcal{M}' \subseteq \mathcal{M}_{\mathrm{dt}}$, let $U \in [0, 1]$, let $S' \in [0, 1]$ be a proper scoring rule in the classical sense and let there be a $\Delta > 0$ such that $\min \{|E_M(U(a, Y) \mid a) - E_M(U(a', Y) \mid a')| : a \neq a'\} > \Delta$ for all $M \in \mathcal{M}'$, then $S(F, a, y) := U(a_F, y) + \Delta \cdot S'(F(Y \mid A = a_F), y)$ is proper and incentive-compatible with $U$. If $S'$ is strictly proper in the classical sense, then $S$ is observationally strictly proper.*

## 5 Eliciting correct forecasts with estimates of the performative divergence

Given a scoring rule $S : \mathcal{P}_{\mathcal{Y}} \times \mathcal{Y} \to \mathbb{R}$, forecast $F \in \mathcal{P}_{\mathcal{Y}}$ and distribution $P \in \mathcal{P}_{\mathcal{Y}}$, the generalised entropy is defined as $H(P) := -\overline{S}(P, P)$ and the generalised divergence as $D(F, P) := -H(P) -$

---

[12]We use the term 'incentive compatibility' different than Chen et al. (2011): they let it refer to strict properness of a conditional scoring rule, which implies our notion of incentive compatibility.

$\overline{S}(F,P) = \int \left(S(P,y) - S(F,y)\right) P(\mathrm{d}y)$, following Grünwald and Dawid (2004). If $S$ is strictly proper, the entropy is the maximal attainable expected score – see also Example 4, where the forecaster manipulates towards easier problems, i.e. problems with lower entropy. Recall the examples of classical scoring rules from the beginning of Section 3: the Bregman score induces the Bregman divergence, the log-score induces the Shannon entropy and Kullback-Leibler divergence, and the (negative) expected Brier score equals the mean square error $-\overline{S}(F,P) = E_P[(f-y)^2] = (p-f)^2 + p(1-p)$, which indeed decomposes into the variance $H(P) = p(1-p)$ and the bias term $D(F,P) = (p-f)^2$. The entropy of the energy score is given by $H(P) = \frac{1}{2}\mathbb{E}_{Y,Y'\sim P}\|Y-Y'\|$, and the divergence by $D(F,P) = \mathbb{E}_{Y\sim F,Y'\sim P}\|Y-Y'\| - \frac{1}{2}\mathbb{E}_{Y,Y'\sim P}\|Y-Y'\| - \frac{1}{2}\mathbb{E}_{Y,Y'\sim F}\|Y-Y'\|$. Definitions of generalised entropy and generalised divergence for non-performative conditional forecasts are given in Appendix C.

In the classical setting, maximizing a proper score is equivalent to minimising the divergence: the correct forecast attains zero divergence (Grünwald and Dawid, 2004; Gneiting and Raftery, 2007). Inspired by this, we aim at proper scoring of forecasts with the performative divergence. To this end, we show that the performative divergence is positive and optimised at observationally correct forecasts (Theorem 5). We subsequently suggest how the principal can calculate the divergence (Corollary 1 and Lemma 1), which is a non-trivial problem since the causal mechanism $M$ is unknown to her.

Indeed, for marginal performative forecasts, we see that the deviation of the forecast $F$ from the induced distribution $P_M(Y \,|\, \mathrm{do}(F))$, given by the *performative divergence* $D_\mathrm{p}(F,M) := D(F, P_M(Y \,|\, \mathrm{do}(F)))$ is also positive. We have $D_\mathrm{p}(F,M) = 0$ if $F$ is correct for $M$, and this holds with equivalence if the divergence is induced by a *strictly* proper scoring rule. We suitably extend this to conditional performative forecasts as follows:

**Definition 7.** Given a scoring rule $S$, conditional forecast $F(Y \,|\, A) \in \mathcal{P}_{\mathcal{A}\to\mathcal{Y}}$ and causal model $M \in \mathcal{M}_\mathrm{pc}$, the *performative divergence* is defined as

$$D_\mathrm{pc}(F,M) := \int \left(S(P_M(Y \,|\, A=a, \mathrm{do}(F)), y) - S(F(Y \,|\, A=a), y)\right) P_M(\mathrm{d}a, \mathrm{d}y \,|\, \mathrm{do}(F)).$$

With *performative entropy* $H_\mathrm{pc}(M \,|\, \mathrm{do}(F)) := \int H(P_M(Y \,|\, A=a, \mathrm{do}(F))) P(\mathrm{d}a \,|\, \mathrm{do}(F))$, the performative divergence can be written as $D_\mathrm{pc}(F,M) = -H_\mathrm{pc}(M \,|\, \mathrm{do}(F)) - \overline{S}_\mathrm{pc}(F(Y \,|\, A), M)$.

Although the divergence cannot be written as the expected value of a scoring rule, we use the same nomenclature of Definition 5 and refer to the divergence as (observationally/counterfactually) (strictly) proper, keeping in mind that the divergence is *minimised* at correct forecasts, instead of *maximised*.

**Theorem 5.** *Let $\mathcal{M}' \subseteq \mathcal{M}_\mathrm{pc}$ be such that for every $M \in \mathcal{M}_\mathrm{pc}$ there exists an observationally correct forecast for $Y|A$. If $S$ is proper in the classical sense, then the divergence $D_\mathrm{pc}$ is positive and proper. If $S$ is strictly proper in the classical sense, then $D_\mathrm{pc}$ is observationally strictly proper.*

Notably, Theorem 5 does not require the target to be forecast-invariant. Also note that full support of $P_M(A \,|\, \mathrm{do}(F))$ implies strict properness of the divergence.

When scoring the forecast directly with a conditional scoring rule, this is straightforward for the principal to carry out: upon receiving a conditional forecast $F(Y \,|\, A)$ and observing $a$ and $y$, the resulting score is $S(F, a, y)$. The forecaster picks the forecast which optimises the expected value $\overline{S}_\mathrm{pc}(F, M)$. When scoring with the performative divergence however, upon receiving the forecast $F(Y \,|\, A)$ and observing $a$ and $y$, she should score with $S(P_M(Y \,|\, A=a, \mathrm{do}(F)), y) - S(F(Y \,|\, A=a), y)$. This either requires access to the distribution $P_M(Y \,|\, A=a, \mathrm{do}(F))$ (in which case forecasting is redundant), or more practically, it requires the principal to estimate the entropy, i.e. the expectation of the first term. More generally, the principal can estimate the divergence. If the employed estimator $\hat{D}_\mathrm{pc}$ is unbiased, the forecaster optimizes $\mathbb{E}_M[\hat{D}_\mathrm{pc} \,|\, \mathrm{do}(F)] = D_\mathrm{pc}(F,M)$, which renders the scoring method observationally strictly proper by Theorem 5.

**Corollary 1.** *Let $\mathcal{M}' \subseteq \mathcal{M}_\mathrm{pc}$ be such that for every $M \in \mathcal{M}^\mathrm{pc}$ there exists an observationally correct forecast for $Y|A$, let $S$ be proper in the classical sense, and let $(A_1, Y_1), ..., (A_n, Y_n) \sim P_M(A, Y \,|\, \mathrm{do}(F))$. Any unbiased estimator $\hat{D}_\mathrm{pc}(A_1, ..., Y_n)$ of the performative divergence $D_\mathrm{pc}(F,M)$ is proper, for any sample size $n \in \mathbb{N}$ such that $\hat{D}_\mathrm{pc}$ is well-defined. If $S$ is strictly proper in the classical sense, then $\hat{D}_\mathrm{pc}$ is observationally strictly proper.*

Notably, the variance of the estimator and the sample size are irrelevant for the properness of the method. To demonstrate the applicability of this result, we show that for finite $\mathcal{A}$ and binary or

continuous $Y$, there are strictly proper classical scoring rules with a corresponding unbiased estimator for the performative divergence.

**Lemma 1.** *Consider the setting of Corollary 1. For binary $Y$ and $S$ the Brier score, the estimator*

$$\hat{D}_{\mathrm{pc}}^{Brier} := \frac{1}{n} \sum_{i=1}^{n} (\hat{p}_{a_i} - f_{a_i})^2 - \frac{\hat{p}_{a_i}(1 - \hat{p}_{a_i})}{n_{a_i} - 1}$$

*is unbiased, where we write $n_a := \sum_{i=1}^{n} \mathbb{1}\{a_i = a\}$, $\hat{p}_a := \frac{1}{n_a} \sum_{i=1}^{n} y_i \mathbb{1}\{a_i = a\}$ and $f_a := F(Y = 1 \,|\, A = a)$. For $Y \in \mathbb{R}^n$ with integrable norm and $S$ the energy score, the estimator*

$$\hat{D}_{\mathrm{pc}}^{energy} := \frac{1}{n} \sum_{i=1}^{n} \mathbb{E}_{Y \sim F_{a_i}} \left[ \|Y - y_i\| \right] - \frac{1}{2} \frac{\sum_{\substack{j \neq i \\ a_j = a_i}} \|y_i - y_j\|}{n_{a_i} - 1} - \frac{1}{2} \mathbb{E}_{Y,Y' \sim F_{a_i}} \left[ \|Y - Y'\| \right]$$

*is unbiased.*

Note that both estimators require $n_a \geq 2$ for all $a \in \mathcal{A}$. The Brier score is strictly proper, and the energy score is strictly proper with respect to the class of distributions with finite first moment (Gneiting and Raftery, 2007). The estimator for the energy score requires an evaluation of the expectations, either analytically or via an (unbiased) sampling scheme.

Instead of making a scoring rule proper by subtracting the estimated entropy, one can multiply a classical scoring rule $S$ with estimates of the *inverse probability weights* (IPWs) $1/P_M(A = a \,|\, \mathrm{do}(F))$ and properly score with $S_{\mathrm{IPW}}(F, a, y) := S(F(Y \,|\, A = a), y)/P_M(A = a \,|\, \mathrm{do}(F))$, as proposed by Chen et al. (2011). In Appendix D we prove for completeness that if $P_M(A \,|\, \mathrm{do}(F))$ has full support and $S$ is a (strictly) proper scoring rule in the classical sense, then IPW score $S_{\mathrm{IPW}}$ is (strictly) proper in the performative sense. For properness of the IPW score the principal either requires knowledge of $P_M(A \,|\, \mathrm{do}(F))$, or must be able to estimate it with an unbiased estimator. It is unclear whether such an unbiased estimators exist. In Appendix G we provide an empirical analysis of unbiased estimates of the divergence on synthetic data, and we compare with biased plugin estimators of the divergence and IPW score.

## 6 Parameter estimation based on scoring rules

In the previous sections we have taken the viewpoint of the principal by addressing how to elicit correct forecasts via proper scoring rules, assuming that the forecaster picks a forecast which maximises the expected score. In this section we switch to the viewpoint of the forecaster who aims at estimating the parameter of the forecasting model from data. We show that if $Y \,|\, A$ is forecast-invariant and the parameter of the conditional forecast is estimated using the divergence related to a strictly proper scoring rule, then this yields a correct forecast which is immediately performatively stable, performatively optimal, and subsequently maximises the expected score when the forecaster is scored by a principal using any proper scoring method.

Formally, consider parametrised forecasts $\{F_\theta : \theta \in \Theta\}$ for some parameter space $\Theta \subseteq \mathbb{R}^d$. If $F_\theta$ does *not* affect $Y$ and one has data $(y_1, ..., y_n)$ with empirical distribution $\hat{P}_n$, one can employ scoring rules for parameter estimation by minimising the divergence $\hat{\theta} = \arg\min_\theta D(F_\theta, \hat{P})$, which is equivalent to minimising the empirical negative score $\hat{\theta} = \arg\min_\theta \sum_{i=1}^{n} -S(F_\theta, y_i)$ which satisfies the estimating equation $\sum_{i=1}^{n} \nabla_\theta S(F_\theta, y_i) = 0$, and therefore defines an $M$-estimator (Dawid and Musio, 2014), which (under regularity conditions) is a consistent estimator for a *correct parameter* $\theta^*$, i.e. a parameter such that the forecast $P_{\theta^*}$ is observationally correct. For the log-score this amounts to maximum likelihood estimation. In machine learning, a popular scheme is *score matching* with the *Hyvärinen score*, amounting to the minimisation of the related divergence $D(F, M) = \int_{\mathcal{Y}} |\nabla_y p(y) - \nabla_y f_\theta(y)|^2 \mathrm{d}y$ (Hyvärinen, 2005). We investigate whether estimation based on scoring rules can also be employed when the forecast is performative, by formulating this question in the framework of Perdomo et al. (2020).

**Definition 8.** Given a loss function $\ell(\theta, a, y)$, the corresponding *performative risk* is defined as $R^{\mathrm{p}}(\theta) := \int \ell(\theta, a, y) P_M(\mathrm{d}a, \mathrm{d}y \,|\, \mathrm{do}(F_\theta))$, and a minimiser $\theta_{PO}$ of the performative risk is referred to as *performatively optimal*. The *decoupled performative risk* is defined as $R^{\mathrm{d}}(\theta_{t+1}, \theta_t) := \int \ell(\theta_{t+1}, a, y) P_M(\mathrm{d}a, \mathrm{d}y \,|\, \mathrm{do}(F_{\theta_t}))$, and parameter $\theta_{PS}$ is *performatively stable* if $\theta_{PS} := \arg\min_\theta R^{\mathrm{d}}(\theta, \theta_{PS})$.

The preceding sections concern the *elicitation* of correct forecasts through scoring rules, calculated as the score of a forecast with respect to the distribution that forecast induces. These metrics (expected negative score $R_S^{\mathrm{p}}(\theta) := -\overline{S}_{\mathrm{pc}}(F_\theta, M)$, divergence $R_D^{\mathrm{p}}(\theta) := D_{\mathrm{pc}}(F_\theta, M)$, and analogue definitions for utility score and IPW score) can be interpreted as a generalised type of performative risk, and if these methods are observationally strictly proper, then observational correctness of the forecast is equivalent to performative optimality of the parameter.

For parameter *estimation* the decoupled performative risk is used, which does not correspond to the 'performative' metrics as just mentioned. Instead, they correspond to more classical metrics, with an additional dependency on the previous parameter:

$$R_S^{\mathrm{d}}(\theta_{t+1}, \theta_t) = -\int \overline{S}(F_{\theta_{t+1}}(Y \,|\, A = a), P_M(Y \,|\, A = a, \mathrm{do}(F_{\theta_t}))) P_M(\mathrm{d}a \,|\, \mathrm{do}(F_{\theta_t}))$$

$$R_D^{\mathrm{d}}(\theta_{t+1}, \theta_t) = \int D(F_{\theta_{t+1}}(Y \,|\, A = a), P_M(Y \,|\, A = a, \mathrm{do}(F_{\theta_t}))) P_M(\mathrm{d}a \,|\, \mathrm{do}(F_{\theta_t})),$$

for which we indeed have that $R_S^{\mathrm{d}}(\theta, \theta) = R_S^{\mathrm{p}}(\theta)$ and $R_D^{\mathrm{d}}(\theta, \theta) = R_D^{\mathrm{p}}(\theta)$. If $S$ and $D$ are strictly proper in the classical sense then they are observationally strictly proper for conditional forecasts (see Appendix C) so upon setting $\theta_{t+1} = \arg\min_\theta R_S^{\mathrm{d}}(\theta, \theta_t)$ or equivalently $\theta_{t+1} = \arg\min_\theta R_D^{\mathrm{d}}(\theta, \theta_t)$ we obtain the recurrence relation $F_{\theta_{t+1}} = P_M(Y \,|\, A, \mathrm{do}(F_{\theta_t}))$ (which holds $P_M(A \,|\, \mathrm{do}(F_{\theta_t}))$-almost everywhere), assuming there is no model misspecification. Now, if $\theta_t$ is observationally correct then we obtain $F_{\theta_t} = P_M(Y \,|\, A, \mathrm{do}(F_{\theta_t}))$ and hence $F_{\theta_{t+1}} = F_{\theta_t}$, i.e. performative stability. If $Y|A$ is forecast-invariant then we obtain $F_{\theta_{t+1}} = P_M(Y \,|\, A)$ ($P_M(A \,|\, \mathrm{do}(F_{\theta_t}))$-almost everywhere) in the aforementioned recurrence relation, so if the supports of $P_M(A \,|\, \mathrm{do}(F_{\theta_t}))$ and $P_M(A \,|\, \mathrm{do}(F_{\theta_{t+1}}))$ coincide we see that $F_{\theta_{t+1}}$ is observationally correct. This parameter is in turn performatively optimal for any performative risk for which performative optimality corresponds to correctness of the parameter. We summarise these insights in the following theorem:

**Theorem 6.** *Suppose that for every model $M$ in the class $\mathcal{M}' \subseteq \mathcal{M}_{\mathrm{pc}}$ and every forecast $F \in \mathcal{P}_{\mathcal{A} \to \mathcal{Y}}$ the distribution $P_M(A \,|\, \mathrm{do}(F))$ has full support and that $Y|A$ is forecast-invariant, and let $D$ be the divergence induced by a strictly proper scoring rule. For any parameter $\theta_t$ for conditional forecast $F_{\theta_t}(Y \,|\, A)$, we have that $\theta_{t+1} := \arg\min_\theta R_D^{\mathrm{d}}(\theta, \theta_t)$ yields a correct forecast which is performatively stable and performatively optimal with respect to $R_D(\theta)$.*

Assume that the forecaster will be scored by the principal using a performatively proper scoring method, e.g. with the utility score from Theorem 4, or with an unbiased estimate of the performative divergence as in Corollary 1. If the forecaster has merely estimated her parameter using the procedure in Theorem 6 and does not have an estimate of $P_M(A \,|\, \mathrm{do}(F))$, she is unable to evaluate (and hence optimise) her expected score directly. However, since the expected score is optimised by the correct forecast, the procedure from Theorem 6 optimises her expected score.

**Corollary 2.** *Consider the setting of Theorem 6. If the forecaster is scored with a strictly proper scoring rule, then the parameter $\theta_{t+1} := \arg\min_\theta R_D^{\mathrm{d}}(\theta, \theta_t)$ is an optimiser of the expected score.*

## 7 Discussion

We develop a theory of performative forecasting that describes when and how making predictions can influence the very outcomes being predicted, showing that classical scoring rules need not be proper nor compatible with outcome-optimisation under performativity, and we propose new utility-based and divergence-based scoring methods that incentivise correct forecasting. We further prove that these methods yield stable and optimal parameter estimates under repeated retraining. We hope that our impossibility result and our prominent emphasis on incentive compatibility instill caution when scoring rules are applied in practice, and that the potential solutions inspire further research on reliable and accurate performative predictions.

The provided proper scoring methods depend on a couple of restrictive assumptions which should be scrutinised in practise, and which open up new paths for further research. First, the assumption of forecast-invariance may be hard to motivate in practise. We also assume a discrete sample space $\mathcal{A}$; extension to the continuous case would be of interest. We conjecture that our theory for probabilistic forecasts can be extended to point forecasts (or actually, various functionals of $P_M(Y \,|\, A, \mathrm{do}(F))$) through *consistent scoring functions* (Lambert et al., 2008; Gneiting, 2011; Holzmann and Eulert, 2014).

## Acknowledgments and Disclosure of Funding

This work is supported by Booking.com. We thank Jason Hartline for helpful discussions, and Tilmann Gneiting, Sourbh Bhadane and anonymous reviewers for helpful remarks.

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

# A Introduction to causal models

The causal models that we consider can be interpreted as causal Bayesian networks or structural causal models. For completeness, we briefly present both modelling frameworks. This material can largely be found in Pearl (2009); Bongers et al. (2021); Forré and Mooij (2025); Boeken et al. (2025).

## A.1 Graphs

A *directed acyclic graph* (DAG) is a tuple $G = (V, E)$ with $V$ a finite set of vertices and $E \subset V \times V$ a set of directed edges such that there are no directed cycles. An *acyclic directed mixed graph* (ADMG) is a tuple $G = (V, E, L)$ with vertices $V$, directed edges $E$, and bidirected edges $L \subset V \times V$ such that there are no directed cycles.

Given DAG $G$ with path $\pi = a \mathbin{*\!\!-\!\!*} ... \mathbin{*\!\!-\!\!*} b$, a *collider* is a vertex $v$ with $... \to v \leftarrow ...$ in $\pi$. For sets of vertices $A, B, C \subseteq V$ we say that $A$ and $B$ are *d-separated* given $C$, written $A \perp^d_G B \mid C$, if for every path $\pi = a \mathbin{*\!\!-\!\!*} ... \mathbin{*\!\!-\!\!*} b$ between every $a \in A$ and $b \in B$, there is a collider on $\pi$ that is not an ancestor of $C$, or if there is a non-collider on $\pi$ in $C$. The sets $A$ and $B$ are *d-connected* given $C$ if they are not *d*-separated, written $A \not\perp^d_G B \mid C$.

Given a DAG $G$ over $V \cup W$, the *latent projection* of $G$ onto $V$ is the ADMG $G_V$ with vertices $V$, directed edges $a \to b$ if there is a path $a \to w_1 \to ... \to w_n \to b$ in $G$ with $w_i \in W$ for all $i = 1, ..., n$ (if any), and bi-directed edges $a \leftrightarrow b$ if there is a bifurcation $a \leftarrow w_1 \leftarrow ... \leftarrow w_k \to ... \to w_n \to b$ in $G$ with $w_i \in W$ for all $i = 1, ..., n$ (Verma, 1993).

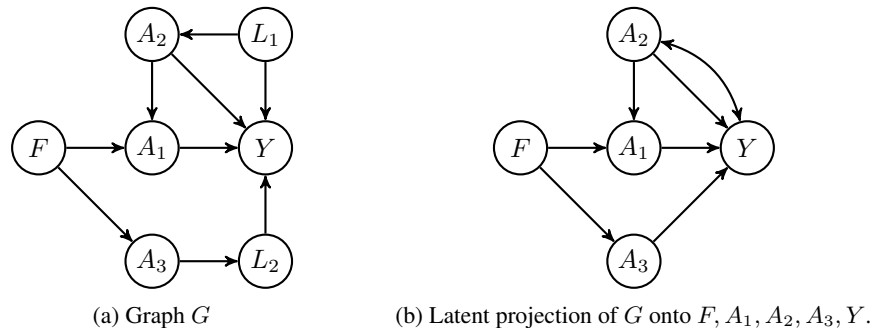

(a) Graph $G$      (b) Latent projection of $G$ onto $F, A_1, A_2, A_3, Y$.

Figure 3: Example of DAG $G$ and a latent projection.

For an ADMG $G$ over $V$, a set of variables $A_1, ..., A_n \subseteq V$ can be *merged* into a single variable $A \notin V$ by removing from $G$ all vertices $A_i$, adding a vertex $A$, and for every $v \in V \setminus \{A_1, ..., A_n\}$ add the edge $v \to A / v \leftrightarrow V / A \to v$ if there is an $i \in \{1, ..., n\}$ such that $v \to A_i / v \leftrightarrow A_i / A_i \to v$ in $G$.

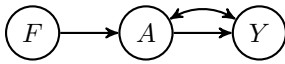

Figure 4: Merging of variables $A_1, A_2, A_3$ of the graph in Figure 3b to a single vertex $A$.

In Figure 3a, we see that $F \perp^d_G Y \mid A_1, A_2, A_3$, so defining $A := (A_1, A_2, A_3)$ we indeed have forecast invariance of $Y|A$ (see Theorem 1). Note that testing for forecast-invariance in the original graph (where the set of variables $A$ is not merged) is stronger than testing it after the merging of $A$: we have $F \perp^d Y \mid A_1, A_2, A_3$ in Figure 3a and equivalently in Figure 3b, but we have $F \not\perp^d Y \mid A$ in Figure 4.

## A.2 Bayesian network

Given a DAG $G$, writing $\mathrm{pa}(v)$ for the set of parents of $v$ in $G$, a *Bayesian network over $G$* is a tuple of Markov kernels $(P(X_v \mid X_{\mathrm{pa}(v)}))_{v \in V}$. We refer to the joint distribution $P(X_V) = \bigotimes_{v \in V} P(X_v \mid X_{\mathrm{pa}(v)})$ as the *observational distribution*.

Given a Bayesian network and a subset of variables $T \subseteq V$ and values $x_T \in \mathcal{X}_T$, the intervened Bayesian network is given by the set of Markov kernels

$$\tilde{P}(X_v \mid X_{\mathrm{pa}(v)}) = \begin{cases} \delta_{x_v}(X_v) & \text{if } v \in T \\ P(X_v \mid X_{\mathrm{pa}(v)}) & \text{if } v \notin T. \end{cases}$$

This gives rise to the interventional distribution $P(X_V \mid \mathrm{do}(X_T = x_t)) := \bigotimes_{v \in V} \tilde{P}(X_v \mid X_{\mathrm{pa}(v)})$, and the graph of the intervened Bayesian network is the graph $G$ without any edges into any of the variables $v \in T$. In the main paper we slightly deviate from this notation by writing e.g. $P_M(Y \mid \mathrm{do}(F))$ to mean $P_M(X_Y \mid \mathrm{do}(X_F = x_F))$, so only for $F$ we make the exception to always indicate a value $x_F$ instead of the random variable $X_F$.

## A.3 Structural Causal Models

Alternatively, the causal models can be interpreted as a *structural causal model* (SCM), which formally consist of a tuple $(V, W, \mathcal{X}, f, P)$ where

- $V$ is the index set of endogenous variables
- $W$ is the index set of exogenous variables
- $\mathcal{X} = \prod_{v \in V} \mathcal{X}_v \times \prod_{w \in W} \mathcal{X}_w$ is a product of Polish spaces (e.g. $\{0,1\}, \mathbb{Z}, \mathbb{R}$)
- $f_V : \mathcal{X}_V \times \mathcal{X}_W \to \mathcal{X}_V$ is a set of *structural equations*
- $P(X_W) = \bigotimes_{w \in W} P(X_w)$ is a product of Borel probability measures.

We call $j \in V \cup W$ a *parent* of $i \in V$ if the structural equation $f_i$ truly depends on $x_j$, i.e. if there is no measurable map $\tilde{f}_i : \mathcal{X}_{V \setminus \{j\}} \times \mathcal{X}_{W \setminus \{j\}} \to \mathcal{X}_i$ such that for $P(X_W)$-almost all $x_W$ and all $x_V$ we have

$$x_i = f_i(x_V, x_W) \iff x_i = \tilde{f}_i(x_{V \setminus \{j\}}, x_{W \setminus \{j\}}).$$

The *augmented graph* of an SCM $M$ is a directed graph $G^a(M)$ with vertices $V \cup W$ and for all $j \in V \cup W, i \in V$ an edge $j \to i$ if $j$ is a parent of $i$. The *graph* $G(M)$ of an SCM $M$ is the latent projection of $G^a(M)$ onto $V$. We call an SCM $M$ *acyclic* if $G^a(M)$ (or equivalently, $G(M)$) is acyclic.

Given an SCM $M = (V, W, \mathcal{X}, f, P)$ and a subset of variables $T \subseteq V$ and value $x_T \in \mathcal{X}_T$, the intervened SCM $M_{\mathrm{do}(X_T = x_T)} = (V, W, \mathcal{X}, \tilde{f}, P)$ has for every $v \in V$ the structural equation

$$\tilde{f}_v = \begin{cases} x_v & \text{if } v \in T \\ f_v(x_V, x_W) & \text{if } v \notin T. \end{cases}$$

Given a subset $T \subseteq V$ and writing $O := V \setminus T$, a *solution function with respect to $O$* is a measurable map $g_O : \mathcal{X}_T \times \mathcal{X}_W \to \mathcal{X}_O$ such that $P(X_W)$-almost all $x_W \in \mathcal{X}_W$ and for all $x_T \in \mathcal{X}_T$ we have $g_O(x_T, x_W) = f_O(x_T, g_O(x_T, x_W), x_W)$. Given a random variable $X_W$ with distribution $P(X_W)$, the random variable $X_V := g_V(X_W)$ defines the *observational distribution* $P_M(X_V)$, and the random variable $X_O := g_O(x_T, X_W)$ defines the *interventional distribution* $P_M(X_O \mid \mathrm{do}(X_T = x_T))$. If $M$ is acyclic, $g_O$ can always be obtained by recursive substitution of the structural equations of $M_{\mathrm{do}(X_T = x_T)}$.

## A.4 Markov property and faithfulness

For any Bayesian network or acyclic SCM with DAG $G$ and observational distribution $P$ the *global Markov property* holds:

$$A \perp^d_G B \mid C \implies X_A \underset{P}{\perp\!\!\!\perp} X_B \mid X_C$$

for all $A, B, C \subseteq V$. A Bayesian network is called *faithful* if for all $A, B, C \subseteq V$ we have

$$A \overset{d}{\underset{G}{\not\perp}} B \mid C \implies X_A \underset{P}{\not\perp\!\!\!\perp} X_B \mid X_C,$$

see also Boeken et al. (2025).

We refer to a distribution $P(X_V)$ as *compatible* with ADMG $G$ with vertices $V$ if the Markov property holds.

# B  Proofs

## B.1  Section 2

**Theorem 1.** *Let $\mathcal{M}'_G \subseteq \mathcal{M}_{\mathrm{pc}}$ be the set of models compatible[13] with causal graph $G$, then the following are equivalent:*

    *i) For every $M \in \mathcal{M}'_G$ there exists an observationally correct forecast of $Y|A$;*

    *ii) For every $M \in \mathcal{M}'_G$ there exists a correct forecast of $Y|A$;*

    *iii) $Y|A$ is forecast-invariant for all $M \in \mathcal{M}'_G$;*

    *iv) $Y \perp^d_G F \mid A$.[14]*

*Proof.* iv) $\implies$ iii): If $F \perp^d_G Y \mid A$ then for every $M \in \mathcal{M}'_G$ we have $P_M(Y \mid A, \mathrm{do}(F)) = P_M(Y \mid A)$ by the third rule of the do-calculus (Pearl, 2009).

iii) $\implies$ ii): If $P_M(Y \mid A, \mathrm{do}(F)) = P_M(Y \mid A)$ for every $M \in \mathcal{M}'_G$, then $P_M(Y \mid A)$ is a correct forecast.

ii) $\implies$ i): This follows immediately from Definition 2.

i) $\implies$ iv): We prove the contrapositive. If $F \not\perp^d_G Y \mid A$, then there must be a path $F \to V_1 \to ... \to V_n \to Y$ in $G$ with $V_i \notin A$ for all $i = 1, ..., n$. Let $P_0, P_1 \in \mathcal{P}_\mathcal{Y}$ with $P_0 \neq P_1$. Set

$$P_M(V_1 \mid \mathrm{do}(F)) = \begin{cases} \delta_0 & \text{if } F(Y \mid A = a) = P_1 \text{ for some } a \in \mathcal{A} \\ \delta_1 & \text{otherwise} \end{cases}$$

$$P_M(V_{i+1} \mid \mathrm{do}(V_i)) = \begin{cases} \delta_0 & \text{if } V_i = 0 \\ \delta_1 & \text{otherwise} \end{cases} \quad \text{for all } i = 1, ..., n-1$$

$$P_M(Y \mid V_n) = \begin{cases} \delta_0 & \text{if } V_n = 0 \\ \delta_1 & \text{otherwise,} \end{cases}$$

and pick for every other variable $Z$ (including the variables in $A$) some independent distribution $P_M(Z)$. Then $M$ is Markov with respect to $G$ so $M \in \mathcal{M}'_G$, and $P_M(Y \mid A, \mathrm{do}(F)) = P_1$ iff $P_1 \notin F(Y \mid A)$, so there is no observationally correct forecast for $M$. $\blacksquare$

## B.2  Section 3

**Theorem 2.** *Let $\mathcal{M}' \subseteq \mathcal{M}_{\mathrm{pc}}$ be the set of models such that $Y|A$ is forecast-invariant, where $P_M(A \mid \mathrm{do}(F))$ has full support for all $F \in \mathcal{P}_\mathcal{Y}$ and all $M \in \mathcal{M}'$, or where $P_M(A \mid \mathrm{do}(F))$ is deterministic for all $M \in \mathcal{M}'$. If $S$ is a scoring rule with respect to $\mathcal{P}_\mathcal{Y}$ such that there are $P_0, P_1 \in \mathcal{P}_\mathcal{Y}$ and forecast $\tilde{F} \in \mathcal{P}_\mathcal{Y}$ with $\tilde{F} \neq P_1$ and $\overline{S}(P_0, P_0) < \overline{S}(\tilde{F}, P_1) < \overline{S}(P_1, P_1)$, then $S$ is not observationally proper for predicting $Y|A$ in $\mathcal{M}'$.*

*Proof.* Let $P_0, P_1 \in \mathcal{P}_\mathcal{Y}$ with $P_0 \neq P_1$, and let $M$ be such that $P_M(Y \mid \mathrm{do}(A = 1)) = P_1$ and $P_M(Y \mid \mathrm{do}(A \neq 1)) = P_0$. For the case where $P_M(A \mid \mathrm{do}(F))$ is deterministic, let

$$P_M(A \mid \mathrm{do}(F)) = \begin{cases} \delta_1 & \text{if } F(Y \mid A = 1) = \tilde{F} \\ \delta_0 & \text{otherwise,} \end{cases}$$

then $Y|A$ is forecast-invariant in $M$ with $F^*(Y \mid A = 1) = P_1$ and $F^*(Y \mid A \neq 1) = P_0$ the correct forecast, and the incorrect forecast $\tilde{F}(Y \mid A = 1) = \tilde{F}$ and $\tilde{F}(Y \mid A \neq 1) = P_0$ has

$$\overline{S}(F^*, M) = \overline{S}(P_0, P_0) < \overline{S}(\tilde{F}, P_1) = \overline{S}(\tilde{F}, M),$$

so $S$ is not proper with respect to $M \in \mathcal{M}'$.

For the case where $P_M(A \mid \mathrm{do}(F))$ has full support, let $Q(A)$ be a probability distribution on $\mathcal{A}$ with full support and such that $Q(A = 1) < \frac{\overline{S}(\tilde{F}, P_1) - \overline{S}(P_0, P_0)}{\overline{S}(P_1, P_1) - \overline{S}(\tilde{F}, P_1)}$; note that this right-hand side is strictly

---

[13]Meaning that the Markov property holds: $A \perp^d_G B \mid C \implies A \perp\!\!\!\perp_{P_M} B \mid C$ for all sets of variables $A, B, C$.

[14]We can have a d-connection $Y \not\perp^d_{G(M)} F \mid A$ but forecast invariance in $M$ due to faithfulness violations after intervention $\mathrm{do}(F)$ (Boeken et al., 2025). However, forecast invariance *for all $M \in \mathcal{M}'_G$* implies $Y \perp^d_G F \mid A$.

positive by assumption. Then the mechanism

$$P_M(A \mid \mathrm{do}(F)) = \begin{cases} \frac{1}{2} \cdot Q + \frac{1}{2} \cdot \delta_1 & \text{if } F(Y \mid A = 1) = \tilde{F} \\ \frac{1}{2} \cdot Q + \frac{1}{2} \cdot \delta_0 & \text{otherwise} \end{cases}$$

has full support for all forecasts $F$ and $Y|A$ is forecast-invariant in $M$, and we have correct forecast $F^*(Y \mid A = 1) = P_1$ and $F^*(Y \mid A \neq 1) = P_0$ and observationally incorrect forecast $\tilde{F}(Y \mid A = 1) = \tilde{F}$ and $\tilde{F}(Y \mid A \neq 1) = P_0$, and we have expected scores

$$2\overline{S}(F^*, M) = (Q(A \neq 1) + 1)\overline{S}(P_0, P_0) + Q(A = 1)\overline{S}(P_1, P_1)$$
$$< Q(A \neq 1)\overline{S}(P_0, P_0) + (Q(A = 1) + 1)\overline{S}(\tilde{F}, P_1) = 2\overline{S}(\tilde{F}, M),$$

so $S$ is not observationally proper. ∎

### B.3   Section 4

**Theorem 3.** *The utility score is incentive-compatible with $U$, and proper if $Y|A$ is forecast-invariant in $\mathcal{M}' \subseteq \mathcal{M}_{\mathrm{dt}}$. If for a given causal graph $G$ one considers the class of models $\mathcal{M}'_G \subseteq \mathcal{M}_{\mathrm{dt}}$ which are consistent with $G$, then forecast invariance of $Y|A$ is* necessary *for the utility score to be proper.*

*Proof.* Since $\overline{S}(F, M) = \int U(a_F, y)P_M(\mathrm{d}y \mid \mathrm{do}(F)) = \int U(a, y)P_M(\mathrm{d}a, \mathrm{d}y \mid \mathrm{do}(F))$ we automatically have

$$\arg\max_F \overline{S}(F, M) = \arg\max_F \int U(a, y)P_M(\mathrm{d}a, \mathrm{d}y \mid \mathrm{do}(F)),$$

so $S$ is incentive compatible with $U$. If $Y|A$ is forecast-invariant, for every $M \in \mathcal{M}'$ the forecast $F(Y \mid A) = P_M(Y \mid A)$ is a correct forecast. By definition of the Bayes act $a_F$ we have $a_{P_M(Y|A)} = \arg\max_a \int U(a, y)P_M(\mathrm{d}y \mid A = a)$ so for $F \in \mathcal{P}_{\mathcal{A} \to \mathcal{Y}}$ we have

$$\overline{S}(F(Y \mid A), M) = \int U(a_F, y)P_M(\mathrm{d}y \mid A = a_F)$$
$$\leq \int U(a_{P_M}, y)P_M(\mathrm{d}y \mid A = a_{P_M}) = \overline{S}(P_M(Y \mid A), M).$$

Since the expected score is maximised at a correct forecast we have that $S$ is proper.
If $Y|A$ is not forecast-invariant in $\mathcal{M}'_G$, then by Theorem 1 there does not exist an observationally correct forecast, so the utility score is not proper. (Note that in the proof of Theorem 1, the counterexample for which no observationally correct forecast exists can easily be altered to let $A = a_F$ be the Bayes act, so that the theorem holds in the class $\mathcal{M}'_G \subseteq \mathcal{M}_{\mathrm{dt}}$.) ∎

**Theorem 4.** *Let $Y|A$ be forecast-invariant in $\mathcal{M}' \subseteq \mathcal{M}_{\mathrm{dt}}$, let $U \in [0, 1]$, let $S' \in [0, 1]$ be a proper scoring rule in the classical sense and let there be a $\Delta > 0$ such that $\min\{|E_M(U(a, Y) \mid a) - E_M(U(a', Y) \mid a')| : a \neq a'\} > \Delta$ for all $M \in \mathcal{M}'$, then $S(F, a, y) := U(a_F, y) + \Delta \cdot S'(F(Y \mid A = a_F), y)$ is proper and incentive-compatible with $U$. If $S'$ is strictly proper in the classical sense, then $S$ is observationally strictly proper.*

*Proof.* We have

$$\overline{S}(F, M) = \int U(a, y)P_M(\mathrm{d}a, \mathrm{d}y \mid \mathrm{do}(F)) + \Delta\overline{S}'(F, M),$$

so if $F^* = \arg\max_F \int U(a, y)P_M(\mathrm{d}a, \mathrm{d}y \mid \mathrm{do}(F))$, then for any other $F$ we have

$$\overline{S}(F^*, M) - \overline{S}(F, M) = \int U(a, y)P_M(\mathrm{d}a, \mathrm{d}y \mid \mathrm{do}(F^*)) - \int U(a, y)P_M(\mathrm{d}a, \mathrm{d}y \mid \mathrm{do}(F))$$
$$+ \Delta(\overline{S}'(F^*, M) - \overline{S}'(F, M))$$
$$> \Delta + \Delta(\overline{S}'(F^*, M) - \overline{S}'(F, M)) \geq 0$$

since $S' \in [0, 1]$, hence

$$\arg\max_F \overline{S}(F, M) = \arg\max_F \int U(a, y) P_M(\mathrm{d}a, \mathrm{d}y \,|\, \mathrm{do}(F)),$$

so $S$ is incentive compatible with $U$. If $Y|A$ is forecast-invariant, then for every $M \in \mathcal{M}'$ the forecast $F^*(Y \,|\, A) = P_M(Y \,|\, A)$ is a correct forecast. Note that any optimiser $F$ of $\overline{S}(F, M)$ must induce the action $a_F = a^* := a_{F^*}$. If $F(Y \,|\, A = a^*) \neq F^*(Y \,|\, A = a^*)$, then $\overline{S}'(F, M) < \overline{S}'(F^*, M)$ so also $\overline{S}(F, M) < \overline{S}(F', M)$, hence $S$ is observationally strictly proper. ∎

## B.4 Section 5

**Theorem 5.** *Let $\mathcal{M}' \subseteq \mathcal{M}_{\mathrm{pc}}$ be such that for every $M \in \mathcal{M}_{\mathrm{pc}}$ there exists an observationally correct forecast for $Y|A$. If $S$ is proper in the classical sense, then the divergence $D_{\mathrm{pc}}$ is positive and proper. If $S$ is strictly proper in the classical sense, then $D_{\mathrm{pc}}$ is observationally strictly proper.*

*Proof.* If $S$ is proper in the classical sense, we have for every $F(Y \,|\, A) \in \mathcal{P}_{\mathcal{A} \to \mathcal{Y}}$ and every $a \in \mathcal{A}$ that

$$\overline{S}(F(Y \,|\, A = a), P_M(Y \,|\, A = a, \mathrm{do}(F))) \leq \overline{S}(P_M(Y \,|\, A = a, \mathrm{do}(F)), P_M(Y \,|\, A = a, \mathrm{do}(F)))$$
$$=: H(P_M(Y \,|\, A = a, \mathrm{do}(F)))$$

so the divergence is positive:

$$D_{\mathrm{pc}}(F, M) = \int \left( S(P_M(Y \,|\, A = a, \mathrm{do}(F)), y) - S(F(Y \,|\, A = a), y) \right) P_M(\mathrm{d}a, \mathrm{d}y \,|\, \mathrm{do}(F))$$

$$= \int \left( H(P_M(Y \,|\, A = a, \mathrm{do}(F))) - \overline{S}(F(Y \,|\, A = a), P_M(Y \,|\, A = a, \mathrm{do}(F))) \right) P_M(\mathrm{d}a \,|\, \mathrm{do}(F))$$

$$\geq 0.$$
(1)

If $F^*(Y \,|\, A)$ is correct for $M$, then $F^*(Y \,|\, A = a) = P_M(Y \,|\, A = a, \mathrm{do}(F^*))$ for all $a \in \mathcal{A}$ such that $P_M(A = a \,|\, \mathrm{do}(F^*)) > 0$, so for every such $a$ we have

$$\overline{S}(F^*(Y \,|\, A = a), P_M(Y \,|\, A = a, \mathrm{do}(F^*))) = \overline{S}(P_M(Y \,|\, A = a, \mathrm{do}(F^*)), P_M(Y \,|\, A = a, \mathrm{do}(F^*)))$$
$$= H(P_M(Y \,|\, A = a, \mathrm{do}(F)))$$

and by plugging this into (1) we get $D_{\mathrm{pc}}(F^*, M) = 0$, so $-D_{\mathrm{pc}}$ is indeed proper. If $S$ is strictly proper, then for any $F(Y \,|\, A = a) \neq P_M(Y \,|\, A = a, \mathrm{do}(F))$ for some $a$ such that $P_M(A = a \,|\, \mathrm{do}(F)) > 0$ then we have

$$H(P_M(Y \,|\, A = a, \mathrm{do}(F))) - \overline{S}(F(Y \,|\, A = a), P_M(Y \,|\, A = a, \mathrm{do}(F^*))) > 0$$

and hence $D_{\mathrm{pc}}(F(Y \,|\, A), M) > 0$, so $-D$ is indeed strictly observationally proper. ∎

**Corollary 1.** *Let $\mathcal{M}' \subseteq \mathcal{M}_{\mathrm{pc}}$ be such that for every $M \in \mathcal{M}^{\mathrm{pc}}$ there exists an observationally correct forecast for $Y|A$, let $S$ be proper in the classical sense, and let $(A_1, Y_1), ..., (A_n, Y_n) \sim P_M(A, Y \,|\, \mathrm{do}(F))$. Any unbiased estimator $\hat{D}_{\mathrm{pc}}(A_1, ..., Y_n)$ of the performative divergence $D_{\mathrm{pc}}(F, M)$ is proper, for any sample size $n \in \mathbb{N}$ such that $\hat{D}_{\mathrm{pc}}$ is well-defined. If $S$ is strictly proper in the classical sense, then $\hat{D}_{\mathrm{pc}}$ is observationally strictly proper.*

*Proof.* By unbiasedness of the estimator we have $\int \hat{D}_{\mathrm{pc}}(A_1, ..., Y_1) \mathrm{d}P_M^n(A, Y) \,|\, \mathrm{do}(F) = D_{\mathrm{pc}}(F, M)$, so the result immediately follows from Theorem 5. ∎

**Lemma 1.** *Consider the setting of Corollary 1. For binary $Y$ and $S$ the Brier score, the estimator*

$$\hat{D}_{\mathrm{pc}}^{Brier} := \frac{1}{n} \sum_{i=1}^{n} (\hat{p}_{a_i} - f_{a_i})^2 - \frac{\hat{p}_{a_i}(1 - \hat{p}_{a_i})}{n_{a_i} - 1}$$

*is unbiased, where we write* $n_a := \sum_{i=1}^n \mathbb{1}\{a_i = a\}$, $\hat{p}_a := \frac{1}{n_a}\sum_{i=1}^n y_i \mathbb{1}\{a_i = a\}$ *and* $f_a := F(Y = 1 \mid A = a)$. *For* $Y \in \mathbb{R}^n$ *with integrable norm and* $S$ *the energy score, the estimator*

$$\hat{D}_{\mathrm{pc}}^{energy} := \frac{1}{n}\sum_{i=1}^n \mathbb{E}_{Y\sim F_{a_i}}\left[\|Y - y_i\|\right] - \frac{1}{2}\frac{\sum_{\substack{j\neq i \\ a_j = a_i}}\|y_i - y_j\|}{n_{a_i} - 1} - \frac{1}{2}\mathbb{E}_{Y,Y'\sim F_{a_i}}\left[\|Y - Y'\|\right]$$

*is unbiased.*

*Proof.* Note that we have $D_{\mathrm{pc}}^{Brier}(F, M) = \sum_{a\in\mathcal{A}}(p_a - f_a)^2 P_M(A = a \mid \mathrm{do}(F))$. By taking the expectation over the data, we have

$$\mathbb{E}\left[\hat{D}_{\mathrm{pc}}^{Brier}\right] = \mathbb{E}\left[\sum_{a\in\mathcal{A}}\frac{n_a}{n}(\hat{p}_{a_i} - F_{a_i})^2 - \frac{\hat{p}_{a_i}(1 - \hat{p}_{a_i})}{n_{a_i} - 1}\right].$$

Writing $p_a := P(Y = 1 \mid A, \mathrm{do}(F))$, for a given $a \in \mathcal{A}$ in the summation and by conditioning on $n_a$ we have $\mathbb{E}[(\hat{p}_a - f_a)^2 \mid n_a] = (p_a - f_a)^2 + \frac{1}{n_a}p_n(1 - p_n)$ and $\mathbb{E}[\hat{p}_a(1 - \hat{p}_a) \mid n_a] = \frac{n_a - 1}{n_a}p_a(1 - p_a)$, and hence $\mathbb{E}[\frac{n_a}{n}(\hat{p}_{a_i} - f_{a_i})^2 - \frac{\hat{p}_{a_i}(1 - \hat{p}_{a_i})}{n_{a_i} - 1} \mid n_a] = \frac{n_a}{n}(p_a - f_a)^2$. Integrating out the $n_a$ we have $\mathbb{E}[\frac{n_a}{n}] = P_M(A \mid \mathrm{do}(F))$ and hence $\mathbb{E}[\hat{D}_{\mathrm{pc}}^{Brier}] = \sum_{a\in\mathcal{A}}(p_a - f_a)^2 P(A \mid \mathrm{do}(F)) = D_{\mathrm{pc}}^{Brier}(F, M)$, which is the desired result.

For the Energy score, the divergence $D_{\mathrm{pc}}^{Energy}$ is equal to

$$\sum_{a\in\mathcal{A}}\left(\mathbb{E}_{\substack{Y\sim F_a \\ Y'\sim P_a}}[\|Y - Y'\|] - \frac{1}{2}\mathbb{E}_{Y,Y'\sim P_a}[\|Y - Y'\|] - \frac{1}{2}\mathbb{E}_{Y,Y'\sim F_a}[\|Y - Y'\|]\right)P_M(A = a \mid \mathrm{do}(F)).$$

Taking the expectation of the first term in the estimator we obtain $\mathbb{E}\left[\frac{1}{n}\sum_{i=1}^n \mathbb{E}_{Y\sim F_{a_i}}[\|Y - y_i\|]\right] = \sum_{a\in\mathcal{A}}\mathbb{E}_{Y\sim F_a, Y'\sim P_a}[\|Y - Y'\|]P_M(A = a \mid \mathrm{do}(F))$. Denoting $I_a := \{i : A_i = a\}$, we see for the second term that

$$\frac{1}{n}\sum_{i=1}^n \frac{1}{2}\frac{\sum_{\substack{j\neq i \\ a_j = a_i}}\|y_i - y_j\|}{n_{a_i} - 1} = \sum_{a\in\mathcal{A}}\frac{1}{2}\left(\frac{1}{n_a(n_a - 1)}\sum_{\substack{i\neq j \\ i,j\in I_a}}\|Y_i - Y_j\|\right)\frac{n_a}{n},$$

where the part between brackets is a standard U-statistic for the norm (also known as *Gini's mean difference*) of independent random variables $Y, Y' \sim P_M(Y \mid A = a, \mathrm{do}(F))$, so the expectation of this term equals $\sum_{a\in\mathcal{A}}\frac{1}{2}\mathbb{E}_{Y,Y'\sim P_a}[\|Y - Y'\|]P_M(A = a \mid \mathrm{do}(F))$ (Székely and Rizzo, 2013). The expected value of the last term clearly equals $\sum_{a\in\mathcal{A}}\frac{1}{2}\mathbb{E}_{Y,Y'\sim F_a}[\|Y - Y'\|]P_M(A = a \mid \mathrm{do}(F))$, so upon combining these results we see that $\mathbb{E}[\hat{D}_{\mathrm{pc}}^{Energy}] = D_{\mathrm{pc}}^{Energy}$. ∎

### B.5 Section 6

**Theorem 6.** *Suppose that for every model $M$ in the class $\mathcal{M}' \subseteq \mathcal{M}_{\mathrm{pc}}$ and every forecast $F \in \mathcal{P}_{\mathcal{A}\to\mathcal{Y}}$ the distribution $P_M(A \mid \mathrm{do}(F))$ has full support and that $Y|A$ is forecast-invariant, and let $D$ be the divergence induced by a strictly proper scoring rule. For any parameter $\theta_t$ for conditional forecast $F_{\theta_t}(Y \mid A)$, we have that $\theta_{t+1} := \arg\min_\theta R_D^{\mathrm{d}}(\theta, \theta_t)$ yields a correct forecast which is performatively stable and performatively optimal with respect to $R_D(\theta)$.*

*Proof.* Since $P_M(A \mid \mathrm{do}(F_{\theta_t}))$ has full support, we have by Theorem 8 that the decoupled performative risk $R_D^{\mathrm{d}}$ is strictly proper, hence we have that $F_{\theta_{t+1}}$ is correct. Similarly we have that $\theta_{t+2} := \arg\min_\theta R_D^{\mathrm{d}}(\theta, \theta_{t+1})$ is correct as well, hence $\theta_{t+1} = \theta_{t+2}$, so $\theta_{t+1}$ is performatively stable. By Theorem 5 we have that the performative risk $R_D$ is strictly proper as well, so $R_D(\theta_{t+1}) = 0$, hence $\theta_{t+1}$ is performatively optimal. ∎

## C   Proper scoring rules for non-performative conditional forecasts

As a special case of Definition 2, given a Markov kernel $P(Y \mid A) \in \mathcal{P}_{\mathcal{A}\to\mathcal{Y}}$ and distribution $P(A) \in \mathcal{P}_{\mathcal{A}}$, the conditional forecast $F(Y \mid A) \in \mathcal{P}_{\mathcal{A}\to\mathcal{Y}}$ is:

- *observationally correct* if $F(Y \mid A = a) = P(Y \mid A = a)$ for all $a \in \mathcal{A}$ such that $P(A = a) > 0$;
- *counterfactually correct* if $F(Y \mid A = a) = P(Y \mid A = a)$ for all $a \in \mathcal{A}$ such that $P(A = a) = 0$;
- *correct* if it is observationally and counterfactually correct.

Given a conditional scoring rule $S : \mathcal{P}_{\mathcal{A} \to \mathcal{Y}} \times \mathcal{A} \to \mathcal{Y} \to \mathbb{R}$, conditional forecast $F(Y \mid A) \in \mathcal{P}_{\mathcal{A} \to \mathcal{Y}}$ and model $(P(Y \mid A), P(A)) \in \mathcal{P}_{\mathcal{A} \to \mathcal{Y}} \times \mathcal{P}_{\mathcal{A}}$, the expected score is

$$\overline{S}_{\mathrm{c}}(F(Y \mid A), P(A, Y)) := \int S(F(Y \mid A), a, y) P(\mathrm{d}a, \mathrm{d}y),$$

where $P(A, Y) := P(Y \mid A) \otimes P(A)$.

**Definition 9.** We call a conditional scoring rule

- *observationally (strictly) proper* relative to $\mathcal{P}_{\mathcal{A} \to \mathcal{Y}} \times \mathcal{P}_{\mathcal{A}}$ if $\overline{S}_{\mathrm{c}}$ is (only) maximised at observationally correct forecasts for all $M \in \mathcal{M}'$;
- *counterfactually (strictly) proper* relative to $\mathcal{P}_{\mathcal{A} \to \mathcal{Y}} \times \mathcal{P}_{\mathcal{A}}$ if $\overline{S}_{\mathrm{c}}$ is (only) maximised at counterfactually correct forecasts for all $M \in \mathcal{M}'$;
- *(strictly) proper* relative to $\mathcal{P}_{\mathcal{A} \to \mathcal{Y}} \times \mathcal{P}_{\mathcal{A}}$ if it is observationally and counterfactually (strictly) proper, that is, if $\overline{S}_{\mathrm{c}}$ is (only) maximised at correct forecasts for all $(P(Y \mid A), P(A)) \in \mathcal{P}_{\mathcal{A} \to \mathcal{Y}} \times \mathcal{P}_{\mathcal{A}}$.

Recall that any scoring rule $S : \mathcal{P}_{\mathcal{Y}} \times \mathcal{Y} \to \mathbb{R}$ induces a conditional scoring rule $S'$ through $S'(F, a, y) := S(F(Y \mid A = a), y)$.

**Theorem 7.** *If $S$ is a proper scoring rule for marginal forecasts with respect to $\mathcal{P}_{\mathcal{Y}}$, then the induced conditional scoring rule is proper with respect to $\mathcal{P}_{\mathcal{A} \to \mathcal{Y}} \times \mathcal{P}_{\mathcal{A}}$. If $S$ is strictly proper, then the induced conditional scoring rule is observationally strictly proper.*

*Proof.* Given $(P(Y \mid A), P(A)) \in \mathcal{P}_{\mathcal{A} \to \mathcal{Y}} \times \mathcal{P}_{\mathcal{A}}$ we have for every $a \in \mathcal{A}$ with $P(A = a) > 0$ that $F(Y \mid A = a), P(Y \mid A = a) \in \mathcal{P}_{\mathcal{Y}}$, and since $S$ is proper in the classical sense we have

$$\overline{S}(F(Y \mid A = a), P(Y \mid A = a)) \leq \overline{S}(P(Y \mid A = a), P(Y \mid A = a)). \tag{2}$$

Integrating with respect to $P(A)$ gives

$$\overline{S}_c(F(Y \mid A), P(A, Y)) = \int \overline{S}(F(Y \mid A = a), P(Y \mid A = a)) P(\mathrm{d}a)$$
$$\leq \int \overline{S}(P(Y \mid A = a), P(Y \mid A = a)) P(\mathrm{d}a) = \overline{S}_c(P(Y \mid A), P), \tag{3}$$

and since $P(Y \mid A)$ is correct, $S$ is proper. If $S$ is strictly proper in the classical sense, then for $F(Y \mid A) \neq P(Y \mid A)$ there is an $a$ with $P(A = a) > 0$ such that $F(Y \mid A = a) \neq P(Y \mid A = a)$, in which case we have a strict inequality in (2), giving a strict inequality in (3), so $S$ is observationally strictly proper. Note that any other $Q(Y \mid A) \in \mathcal{P}_{\mathcal{A} \to \mathcal{Y}}$ such that $Q(Y \mid A = a) = P(Y \mid A = a)$ for all $a \in \mathcal{A}$ with $P(A = a) > 0$ obtains the same expected score but need not be counterfactually correct, so $S$ is not counterfactually strictly proper. ∎

### C.1 Divergence for non-performative conditional forecasts

For a conditional forecast $F(Y \mid A) \in \mathcal{P}_{\mathcal{A} \to \mathcal{Y}}$ and model $(P(Y \mid A), P(A)) \in \mathcal{P}_{\mathcal{A} \to \mathcal{Y}} \times \mathcal{P}_{\mathcal{A}}$, we define the generalised conditional entropy and generalised conditional divergence as

$$H_{\mathrm{c}}(P(A, Y)) := \int H(P(Y \mid A = a)) P(\mathrm{d}a)$$

$$D_{\mathrm{c}}(F(Y \mid A), P(A, Y)) := H_{\mathrm{c}}(P) - \overline{S}_{\mathrm{c}}(F(Y \mid A), P)$$
$$= \int S(P(Y \mid A = a), y) - S(F(Y \mid A = a), y) P(\mathrm{d}a, \mathrm{d}y).$$

For the log-score, the entropy $H_{\mathrm{c}}$ is also known as the *conditional entropy* (Cover and Thomas, 2006).

For the divergence, we obtain a theorem similar to Theorem 7, that the generalised conditional divergence is observationally (strictly) proper if it is induced by a (strictly) proper scoring rule.

**Theorem 8.** *If $S$ is a proper scoring rule, then $D_c$ is proper for conditional forecasts. If $S$ is strictly proper, then $D_c$ observationally strictly proper.*

*Proof.* Given $(P(Y \mid A), P(A)) \in \mathcal{P}_{\mathcal{A} \to \mathcal{Y}} \times \mathcal{P}_{\mathcal{A}}$ and proper scoring rule $S$ we have

$$D_c(F(Y \mid A), P(A, Y)) = \int D(F(Y \mid A = a), P(Y \mid A = a))P(\mathrm{d}a) \geq 0 \qquad (4)$$

and $D_c(F(Y \mid A), P(A, Y)) = 0$, so $D_c$ is proper. If $S$ is strictly proper then for every $a \in \mathcal{A}$ with $P(A = a) > 0$ we have $D(F(Y \mid A = a), P(Y \mid A = a)) = 0$ if and only if $F(Y \mid A = a) = P(Y \mid A = a)$, hence $D_c(F, P) = 0$ only if $F$ is observationally correct. $\blacksquare$

## D  Proper scoring with inverse probability weights

By scoring with divergence, the utilised scoring rule is made performatively proper by subtracting the (estimated) entropy. As alternative approach, one can multiply the score by (estimates of) the *inverse probability weights* (IPWs) $1/P_M(A \mid \mathrm{do}(F))$, and consider the expected *IPW score*:

$$\overline{S}_{\mathrm{IPW}}(F, M) := \int \frac{S(F(Y \mid A = a), y)}{P_M(A = a \mid \mathrm{do}(F))} P(\mathrm{d}a, \mathrm{d}y \mid \mathrm{do}(F))$$

$$= \sum_{a \in \mathcal{A}} \int S(F(Y \mid A = a), y)P(\mathrm{d}y \mid a).$$

Chen et al. (2011) introduced the IPW score in settings where the principal knows $P_M(A)$ (which does not depend on $F$), who can then after every observation $(a, y)$ properly score with $S_{\mathrm{IPW}}(F, a, y) := S(F(Y \mid A = a), y)/P_M(A = a)$. If the distribution $P_M(A \mid \mathrm{do}(F))$ has full support, then this procedure is strictly proper:

**Theorem 9.** *Suppose that for every model $M$ in the class $\mathcal{M}' \subseteq \mathcal{M}_{\mathrm{pc}}$ and every forecast $F \in \mathcal{P}_{\mathcal{A} \to \mathcal{Y}}$ the distribution $P_M(A \mid \mathrm{do}(F))$ has full support, and that $Y | A$ is forecast-invariant in $M \in \mathcal{M}'$. If $S$ is (strictly) proper in the classical sense, then $\overline{S}_{\mathrm{IPW}}$ is (strictly) proper in the performative sense.*

*Proof.* By forecast-invariance of $Y|A$ we have

$$\overline{S}_{\mathrm{IPW}}(F, M) = \int \int S(F(Y \mid A = a), y)P_M(\mathrm{d}y \mid A = a)\mathrm{d}a$$

$$= \int S(F(Y \mid A = a), P_M(Y \mid A = a))\mathrm{d}a,$$

so for every $a \in \mathcal{A}$ the optimum of the integrand lies at the correct forecast $F^*(Y \mid A = a) = P_M(Y \mid A = a)$, and hence $P_M(Y \mid A)$ maximises $S_{\mathrm{IPW}}$, so $\overline{S}_{\mathrm{IPW}}$ is proper. If $S$ is strictly proper, then for every $F$ with $F(Y \mid A = a) \neq F^*(Y \mid A = a)$ for some $a \in \mathcal{A}$ we have $S(F(Y \mid A = a), P_M(Y \mid A = a)) < S(F^*(Y \mid A = a), P_M(Y \mid A = a))$, so $\overline{S}_{\mathrm{IPW}}(F, M) < \overline{S}_{\mathrm{IPW}}(F^*, M)$, hence $\overline{S}_{\mathrm{IPW}}$ is strictly proper. $\blacksquare$

In Section F.3 we provide an example where $\overline{S}_{\mathrm{IPW}}$ is not counterfactually proper when $P_M(A \mid \mathrm{do}(F))$ does not have full support.

A similar principle as for the divergence applies: if the principal does not have access to $P_M(A \mid \mathrm{do}(F))$, an unbiased estimate of the mean IPW score suffices for the method to be (observationally strictly) proper. It is unclear whether unbiased estimators for the mean IPW score exist. In Section G we provide a brief empirical investigation into the consistency of the plugin estimator for the mean IPW score.

## E  Forecasts depending on covariates

### E.1  Causal model including covariates

We extend the results from the main paper to the setting where we let the forecast $F$ depend explicitly on covariates $X$. In the main paper, we use the notation $P_M(Y \mid \mathrm{do}(F))$ such that for

every forecast $F \in \mathcal{P}_{\mathcal{Y}}$ we can meaningfully talk about the distribution of $Y$, given the forecast $F$. If we let $F$ depend on covariates $X$, then intervening on $F$ would make the forecast invariant of $X$, which is not what we intend to do. Therefore, we consider a parameter $\theta \in \Theta \subseteq \mathbb{R}^d$ for some $d$ which specifies a forecasting *model* $F_\theta(Y \mid X) \in \mathcal{P}_{\mathcal{X} \to \mathcal{Y}}$. In this setting, we will instead of $P_M(Y \mid \mathrm{do}(F))$, for each observed $x \in \mathcal{X}$ consider the causal effect of a specific parameter $\theta$ and its induced forecast $F_\theta(Y \mid X = x)$ on $Y$, which we express via $P_M(Y \mid \mathrm{do}(\theta))$, which is equal to $\int P_M(Y \mid \mathrm{do}(\theta, F_\theta(X = x))) P_M(\mathrm{d}x)$ by consistency.

Formally, for marginal forecasts we let a parameter $\theta$ induce the forecasting model $F_\theta(Y \mid X) \in \mathcal{P}_{\mathcal{X} \to \mathcal{Y}}$, where for every observed value $x$ and parameter $\theta$ the reported marginal forecast is $F_\theta(Y \mid X = x) \in \mathcal{P}_{\mathcal{Y}}$. For this setting let $\mathcal{M}_{\mathrm{p+}}$ be the set of causal models whose projection onto $X, \theta, F, Y$ is a subgraph of Figure 5a.

For conditional forecasts, let $\theta$ induce the forecasting model $F_\theta(Y \mid A, X) \in \mathcal{P}_{\mathcal{A} \times \mathcal{X} \to \mathcal{Y}}$, where for every observed value $x$ and parameter $\theta$ the reported conditional forecast is $F_\theta(Y \mid A, X = x) \in \mathcal{P}_{\mathcal{A} \to \mathcal{Y}}$; let $\mathcal{M}_{\mathrm{pc+}}$ be the set of models whose projection onto $X, \theta, F, A, Y$ is a subgraph of Figure 5b.

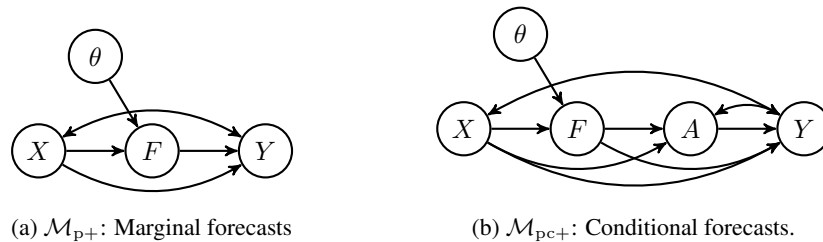

(a) $\mathcal{M}_{\mathrm{p+}}$: Marginal forecasts      (b) $\mathcal{M}_{\mathrm{pc+}}$: Conditional forecasts.

Figure 5: Causal graph of performative forecast depending on covariates $X$.

## E.2    Section 2: Correctness of forecasts

In the following sections, in each definition or theorem, the corresponding definition/theorem number from the main paper is added in parentheses.

**Definition 10** (Definitions 1, 2). For a marginal forecast $F_\theta(Y \mid X = x)$, the parameter is correct for $M \in \mathcal{M}_{\mathrm{pc+}}$ if $F_\theta(Y \mid X = x) = P_M(Y \mid X = x, \mathrm{do}(\theta))$ holds for $P_M(X)$-almost all $x \in \mathcal{X}$. For a conditional forecast $F_\theta(Y \mid A, X = x)$, the parameter $\theta$ is

- *observationally correct* if for $P_M(X)$-almost all $x \in \mathcal{X}$, for all $a \in \mathcal{A}$ such that $P(A = a \mid X = x, \mathrm{do}(\theta)) > 0$ we have $F_\theta(Y \mid A = a, X = x) = P(Y \mid A = a, X = x, \mathrm{do}(\theta))$;

- *counterfactually correct* if for $P_M(X)$-almost all $x \in \mathcal{X}$, for all $a \in \mathcal{A}$ such that $P(A = a \mid X = x, \mathrm{do}(\theta)) = 0$ we have $F_\theta(Y \mid A = a, X = x) = P(Y \mid A = a, X = x, \mathrm{do}(\theta))$;

- *correct* if it is observationally and counterfactually correct.

**Definition 11** (Definition 3). We call the target $Y|A, X$ *forecast-invariant* in $\mathcal{M}' \subseteq \mathcal{M}_{\mathrm{pc+}}$ if the target distribution of $Y|A, X$ does not depend on the forecast, so if $P_M(Y \mid A, X, \mathrm{do}(\theta)) = P_M(Y \mid A, X)$ for all $M \in \mathcal{M}'$.

**Theorem 10** (Theorem 1). *Let $\mathcal{M}'_G \subseteq \mathcal{M}_{\mathrm{pc+}}$ be the set of models compatible with causal graph $G$, then the following are equivalent:*

    i) *For every $M \in \mathcal{M}'_G$ there exists an observationally correct parameter for $Y|A, X$;*

    ii) *For every $M \in \mathcal{M}'_G$ there exists a correct parameter for $Y|A, X$;*

    iii) *$Y|A, X$ is forecast-invariant for all $M \in \mathcal{M}'_G$;*

    iv) *$\theta \perp^d_G Y \mid A, X$.*

*Proof.* The implications iv) $\implies$ iii) $\implies$ ii) $\implies$ i) apply without any change of the proof; for i) $\implies$ iv) one has to also specify an independent distribution $P_M(X)$ to finish the proof. ∎

### E.3 Section 3: Scoring rules

Consider a scoring rule $S : \mathcal{P}_{\mathcal{Y}} \times \mathcal{X} \times \mathcal{Y} \to \mathbb{R}$, where given a marginal forecast $F_\theta(Y \mid X = x)$ and an observed outcome $y$, the forecaster receives the score $S(F_\theta(Y \mid X = x), x, y)$. Considering $M \in \mathcal{M}_{\mathrm{pc+}}$, the expected score is

$$\overline{S}_{\mathrm{p+}}(F_\theta, M) = \int S(F_\theta(Y \mid X = x), x, y) P_M(\mathrm{d}x, \mathrm{d}y \mid \mathrm{do}(\theta)).$$

For a conditional forecast $F_\theta(Y \mid A, X = x)$, conditional scoring rule $S : \mathcal{P}_{\mathcal{A} \to \mathcal{Y}} \times \mathcal{A} \times \mathcal{X} \times \mathcal{Y} \to \mathbb{R}$, observed action $a$ and outcome $y$, the forecaster receives the conditional score $S(F(Y \mid A, X = x), a, x, y)$, and given a model $M \in \mathcal{M}_{\mathrm{pc+}}$ the expected score is

$$\overline{S}_{\mathrm{pc+}}(F_\theta, M) = \int S(F_\theta(Y \mid X = x), a, x, y) P_M(\mathrm{d}a, \mathrm{d}x, \mathrm{d}y \mid \mathrm{do}(\theta)).$$

**Definition 12** (Definition 5). For marginal forecasts, a scoring rule $S$ is *proper* relative to $\mathcal{M}' \subseteq \mathcal{M}_{\mathrm{p+}}$ if $\overline{S}_{\mathrm{p+}}$ is maximised at a correct forecast, and *strictly proper* if all maximisers are correct. We call a conditional scoring rule

- *observationally (strictly) proper* relative to $\mathcal{M}' \subseteq \mathcal{M}_{\mathrm{pc+}}$ if $\overline{S}_{\mathrm{pc+}}$ is (only) maximised at observationally correct forecasts for all $M \in \mathcal{M}'$;
- *counterfactually (strictly) proper* relative to $\mathcal{M}' \subseteq \mathcal{M}_{\mathrm{pc+}}$ if $\overline{S}_{\mathrm{pc+}}$ is (only) maximised at counterfactually correct forecasts for all $M \in \mathcal{M}'$;
- *(strictly) proper* relative to $\mathcal{M}' \subseteq \mathcal{M}_{\mathrm{pc+}}$ if it is observationally and counterfactually (strictly) proper, that is, if $\overline{S}_{\mathrm{pc+}}$ is (only) maximised at correct forecasts for all $M \in \mathcal{M}'$.

**Theorem 11** (Theorem 2). *Let $\mathcal{M}' \subseteq \mathcal{M}_{\mathrm{pc+}}$ be the set of models such that $Y|A, X$ is forecast-invariant, where $P_M(A \mid X = x, \mathrm{do}(\theta))$ has full support for all $\theta \in \Theta$, $P_M(X)$-almost all $x \in \mathcal{X}$ and all $M \in \mathcal{M}'$, or where $P_M(A \mid X = x, \mathrm{do}(\theta))$ is deterministic for all $\theta \in \Theta$, $P_M(X)$-almost all $x \in \mathcal{X}$ and all $M \in \mathcal{M}'$. If $S$ is a scoring rule with respect to $\mathcal{P}_{\mathcal{Y}}$ such that there are $P_0, P_1 \in \mathcal{P}_{\mathcal{Y}}$ and forecast $\tilde{F} \in \mathcal{P}_{\mathcal{Y}}$ with $\tilde{F} \neq P_1$ and $\overline{S}(P_0, P_0) < \overline{S}(\tilde{F}, P_1) < \overline{S}(P_1, P_1)$, then $S$ is not observationally proper for predicting $Y|A, X$ in $\mathcal{M}'$.*

*Proof.* The proof proceeds as in the original proof of Theorem 2 with one adjustment: the constructed counterexample specifies a mechanism for $F \to A \to Y$; specifying any independent distribution $P_M(X)$ completes the proof. ∎

### E.4 Section 4: Decision theory

Given a conditional forecast $F_\theta$, let the agent take an action which maximizes the expectation of a utility $U(a, x, y)$, so we have $A = f_A(F_\theta, x) := \arg\max_{a \in \mathcal{A}} \int U(a, x, y) F_\theta(\mathrm{d}y \mid x, a)$. Let $\mathcal{M}_{\mathrm{dt+}} \subseteq \mathcal{M}_{\mathrm{pc+}}$ be the class of models with $P_M(A \mid X = x, \mathrm{do}(\theta)) = \delta_{a_{\theta, x}}$.

**Definition 13** (Definition 6). Given a utility $U$ a set of causal models $\mathcal{M}' \subseteq \mathcal{M}_{\mathrm{pc+}}$, we call a conditional scoring rule $S$ *incentive compatible* with $U$ if for all $M \in \mathcal{M}'$ we have $\arg\max_\theta \overline{S}_{\mathrm{pc+}}(F_\theta, M) = \arg\max_\theta \int U(a, x, y) P_M(\mathrm{d}a, \mathrm{d}x, \mathrm{d}y \mid \mathrm{do}(\theta))$.

The *utility score* is given by $S(F_\theta, a, x, y) := U(f_A(F_\theta, x), x, y)$.

**Theorem 12** (Theorem 3). *The utility score is incentive-compatible with $U$, and proper if $Y|A, X$ is forecast-invariant in $\mathcal{M}' \subseteq \mathcal{M}_{\mathrm{dt+}}$. If for a given causal graph $G$ one considers the class of models $\mathcal{M}'_G \subseteq \mathcal{M}_{\mathrm{dt+}}$ which are consistent with $G$, then forecast invariance of $Y|A, X$ is* necessary *for the utility score to be proper.*

*Proof.* The proof proceeds analogous to the proof of Theorem 3. ∎

**Theorem 13** (Theorem 4). *If $Y|A$ is forecast-invariant in $\mathcal{M}' \subseteq \mathcal{M}_{\mathrm{dt+}}$, $U \in [0, 1]$, $S' \in [0, 1]$ is a proper scoring rule in the classical sense and $\min\{|E_M(U(a, x, Y) \mid a) - E_M(U(a', x, Y) \mid a')| : a \neq a'\} > \Delta > 0$ for $P_M(X)$-almost all $x \in \mathcal{X}$ and all $M \in \mathcal{M}'$, then $S(F, a, x, y) = U(f_A(F_\theta, x), x, y) + \Delta_x \cdot S'(F(Y \mid A = f_A(F_\theta, x)), y)$ is proper and incentive-compatible with $U$. If $S'$ is strictly proper in the classical sense, then $S$ is observationally strictly proper.*

*Proof.* The proof proceeds analogous to the proof of Theorem 4. ∎

### E.5 Section 5: Divergence

**Definition 14** (Definition 7). Given a scoring rule $S : \mathcal{P}_{\mathcal{Y}} \times \mathcal{Y} \to \mathbb{R}$, conditional forecast $F_\theta(Y \mid A, X = x) \in \mathcal{P}_{\mathcal{A} \to \mathcal{Y}}$ and causal model $M \in \mathcal{M}_{\mathrm{pc}+}$, the *performative divergence* is defined as

$$D_{\mathrm{pc}+}(F_\theta, M) := \int \left( S(P_M(Y \mid a, x, \mathrm{do}(\theta)), y) - S(F_\theta(Y \mid a, x), y) \right) P_M(\mathrm{d}a, \mathrm{d}x, \mathrm{d}y \mid \mathrm{do}(\theta)).$$

With *performative entropy* $H_{\mathrm{pc}+}(M \mid \mathrm{do}(\theta)) := \int H(P_M(Y \mid a, x, \mathrm{do}(\theta))) P(\mathrm{d}a, \mathrm{d}x \mid \mathrm{do}(\theta))$, the performative divergence can be written as $D_{\mathrm{pc}+}(F, M) = -H_{\mathrm{pc}+}(M \mid \mathrm{do}(\theta)) - \overline{S}_{\mathrm{pc}+}(F_\theta(Y \mid A, X), M)$.

**Theorem 14** (Theorem 5). *If for every $M \in \mathcal{M}' \subseteq \mathcal{M}_{\mathrm{pc}+}$ there exists an observationally correct forecast for $Y \mid A, X$ and if $S$ is proper in the classical sense, then the divergence $D_{\mathrm{pc}+}$ is positive and proper. If $S$ is strictly proper in the classical sense, then $D_{\mathrm{pc}+}$ is observationally strictly proper.*

*Proof.* The proof proceeds analogous to the proof of Theorem 5. ∎

**Corollary 3.** *Let for every $M \in \mathcal{M}' \subseteq \mathcal{M}_{\mathrm{pc}+}$ there be an observationally correct forecast for $Y \mid A, X$, let $S$ be proper in the classical sense, and let $(A_1, X_1, Y_1), ..., (A_n, X_n, Y_n) \sim P_M(A, X, Y \mid \mathrm{do}(F_\theta))$. Any unbiased estimator $\hat{D}_{\mathrm{pc}+}(A_1, ..., Y_n)$ of the performative divergence $D_{\mathrm{pc}+}(F_\theta, M)$ is proper, for any sample size $n \in \mathbb{N}$ such that $\hat{D}_{\mathrm{pc}+}$ is well-defined. If $S$ is strictly proper in the classical sense, then $\hat{D}_{\mathrm{pc}+}$ is observationally strictly proper.*

### E.6 Section 6: Parameter estimation

**Definition 15** (Definition 8). Given a loss function $\ell(\theta, a, x, y)$, the corresponding *performative risk* is defined as $R(\theta) := \int \ell(\theta, a, x, y) P_M(\mathrm{d}a, \mathrm{d}x, \mathrm{d}y \mid \mathrm{do}(\theta))$, and a minimiser $\theta_{PO}$ of the performative risk is referred to as *performatively optimal*. The *decoupled performative risk* is defined as $R^{\mathrm{d}}(\theta_{t+1}, \theta_t) := \int \ell(\theta_{t+1}, a, x, y) P_M(\mathrm{d}a, \mathrm{d}x, \mathrm{d}y \mid \mathrm{do}(\theta_t))$, and parameter $\theta_{PS}$ is *performatively stable* if $\theta_{PS} := \arg\min_\theta R^{\mathrm{d}}(\theta, \theta_{PS})$.

We define the performative divergence and decoupled performative divergence as:

$$R_{D+}(\theta) := D_{\mathrm{pc}+}(F_\theta, M)$$

$$R_{D+}^{\mathrm{d}}(\theta_{t+1}, \theta_t) := \int D(F_{\theta_{t+1}}(Y \mid a, x), P_M(Y \mid a, x, \mathrm{do}(\theta_t))) P_M(\mathrm{d}a, \mathrm{d}x \mid \mathrm{do}(\theta_t)).$$

**Theorem 15** (Theorem 6). *Suppose that for every model $M$ in the class $\mathcal{M}' \subseteq \mathcal{M}_{\mathrm{pc}+}$, every parameter $\theta \in \Theta$ and $P_M(X)$-almost all $x \in \mathcal{X}$ the distribution $P_M(A \mid X = x, \mathrm{do}(\theta))$ has full support, and that $Y \mid A, X$ is forecast-invariant in $M \in \mathcal{M}'$, and let $D$ be the divergence induced by a strictly proper scoring rule. For any parameter $\theta_t$ for conditional forecast $F_{\theta_t}(Y \mid A, X = x)$, we have that $\theta_{t+1} := \arg\min_\theta R_{D+}^{\mathrm{d}}(\theta, \theta_t)$ is performatively stable and performatively optimal with respect to $R_{D+}(\theta)$.*

*Proof.* The proof proceeds analogous to the proof of Theorem 6. ∎

## F Examples

Python code to generate the plots of the examples in this section is provided in the supplements. The code generates interactive 3D plots, which can be helpful for visual aid.

For interpreting these plots (see e.g. Figure 6), the reading guide of these plots is as follows. Global optima are indicated with a blue line or dot, the correct forecast is indicated with a green dot. (Strict) Properness means that (only) the correct forecast has the same expected score as the global optima (z-axis). The grey, projections of the true and optimal forecasts can be used to read the score of these forecasts (on the z-axis). Observational/counterfactual (strict) properness means that the global optima are (only) at the correct values of $F(Y \mid A = 0)$ and $F(Y \mid A = 1)$. The bottom projections can be used to read off whether the optimal forecasts coincide with the true forecast. This can

separately be checked for $A = 0$ and $A = 1$. In these examples, at any $F$, the 'counterfactual' value of $A$ (that is, the values $a \in \mathcal{A}$ such that $P_M(A = a \mid \mathrm{do}(F)) = 0$) can be read off by checking the direction in which the expected score at $F$ is *not* piecewise constant.

### F.1 Classical scoring rules which are not performatively proper

The following example is provided in the main paper (Example 4):

**Example 6.** Let $\mathcal{A} = \mathcal{Y} = \{0, 1\}$ and $M$ be such that $Y|A$ is forecast-invariant, with $P_M(Y = 1 \mid A = 0) = 0.5$, $P_M(Y = 1 \mid A = 1) = 0.25$ and the 'decision rule' $A = \arg\max_a F(Y = 1 \mid A = a)$. If the forecaster reports the correct forecast the expected Brier score is $-0.5 \cdot (0.5)^2 - 0.5 \cdot (0.5)^2 = -0.25$, and if the forecaster reports $F(Y = 1 \mid A = 0) = 0.2, F(Y = 1 \mid A = 1) = 0.25$ then the expected score is $-0.25 \cdot (0.75)^2 - 0.75 \cdot (0.25)^2 \approx -0.19$. See also Figure 6: the scoring rule is observationally strictly proper since for the observed value $A = 1$ the score is maximised at a correct forecast, but it is not counterfactually proper since it pays off to misreport for the value $A = 0$ which is not observed. This can also be observed in the figure using the bottom projection of the optimal forecasts and the unique, correct forecast.

The scoring rule in this example is not (counterfactually) proper, but it is observationally strictly proper. As an extension of the self-defeating prophecy, if we pick the harder to predict value of $A$ when the forecast for the easier to predict value of $A$ gets close to being correct, this provides an example where the score is not observationally proper.

**Example 7.** Consider Example 6, but with the decision rule

$$A = \begin{cases} 1 & \text{if } F(Y = 1 \mid A = 0) \leq 0.4 \text{ and } F(Y = 1 \mid A = 1) \geq 0.4 \\ 0 & \text{otherwise.} \end{cases}$$

Since $P_M(Y = 1 \mid A = 1) = 0.25$ is much easier to predict than $P_M(Y = 1 \mid A = 0) = 0.5$, even if an incorrect prediction $F(Y = 1 \mid A = 0) = F(Y = 1 \mid A = 1) = 0.4$ is made such that we observe $A = 1$ the expected score $-0.25 \cdot (0.6)^2 - 0.75 \cdot (0.4)^2 = -0.21$ is higher than for correctly predicting and observing $P_M(Y = 1 \mid A = 0) = 0.5$: if the forecaster reports her true belief the expected score is $-0.5 \cdot (0.5)^2 - 0.5 \cdot (0.5)^2 = -0.25$. Hence, the scoring rule is neither observationally nor counterfactually proper. See the bottom projection in Figure 7a: the optimal forecasts have neither $F(Y \mid A = 0)$ nor $F(Y \mid A = 1)$ correct.

One could expect this problem to be resolved if one considers a stochastic policy for deciding $A$ given $F$. However, the following example shows that full support of $P_M(A \mid \mathrm{do}(F))$ does not resolve this issue:

**Example 8.** Consider the decision rule which is a mixture of the mechanism of Example 7 and a uniform distribution over $A = 0, 1$: with probability $1/3$ there is uniform 'exploration', and with

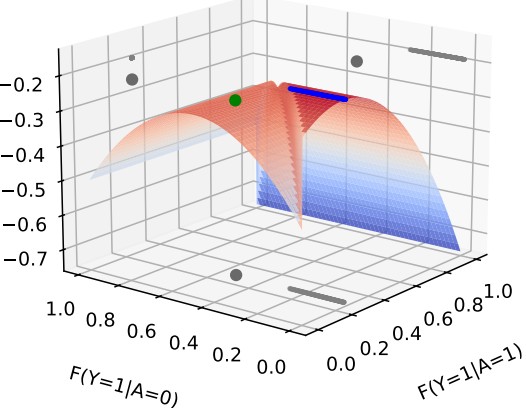

Figure 6: Expected Brier scores $\overline{S}_{\mathrm{pc}}$ for the mechanisms $P_M(A \mid \mathrm{do}(F))$ as defined in Example 6. It is observationally strictly proper, counterfactually improper.

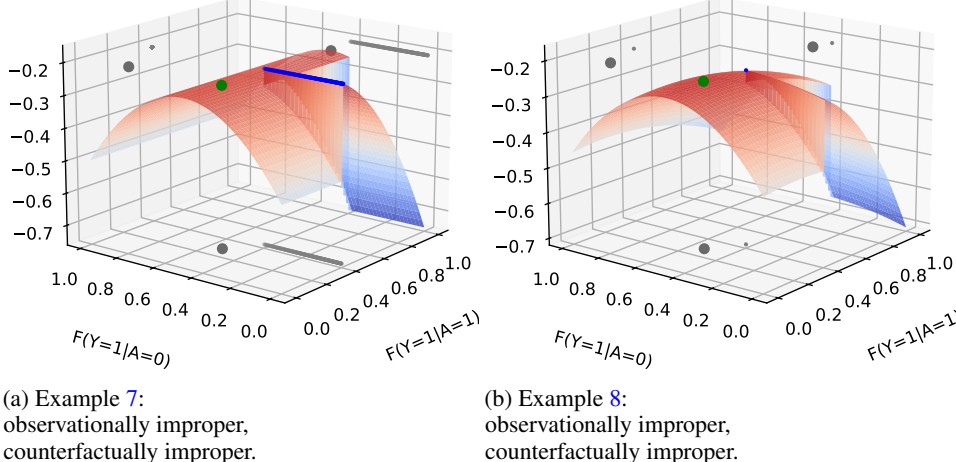

(a) Example 7:
observationally improper,
counterfactually improper.

(b) Example 8:
observationally improper,
counterfactually improper.

Figure 7: Expected Brier score for the self-defeating prophecy, without positivity and with positivity. In both mechanisms expected score is neither observationally proper nor counterfactually proper

probability $2/3$ the decision rule

$$A = \begin{cases} 1 & \text{if } F(Y = 1 \,|\, A = 0) \leq 0.4 \text{ and } F(Y = 1 \,|\, A = 1) \geq 0.4 \\ 0 & \text{otherwise} \end{cases}$$

is used. For this mechanism the correct forecast obtains the expected score of $(\frac{1}{3} \cdot \frac{1}{2} + \frac{2}{3}) \cdot (-0.25) + \frac{1}{3} \cdot \frac{1}{2} \cdot (-0.19) \approx -0.24$, and if the forecaster reports $F(Y = 1 \,|\, A = 0) = F(Y = 1 \,|\, A = 1) = 0.4$ then the expected score is $(\frac{1}{3} \cdot \frac{1}{2}) \cdot (-0.25) + (\frac{1}{3} \cdot \frac{1}{2} + \frac{2}{3}) \cdot (-0.21) \approx -0.22$, so the scoring rule is not observationally proper. See Figure 7b.

## F.2 Incentive-compatible and proper utility scores

**Example 9.** Consider Example 6, interpreted as a forecast being given to an agent with utility $U(a, y) = y$; then indeed $A = a_F = \arg\max_a F(Y = 1 \,|\, A = a)$. The optimal action for the agent is $a^* = 0$, but to maximise her score (with respect to the Brier score, see Figure 6) the forecaster reports $F(Y = 1 \,|\, A = 0) = 0.2$ and $F(Y = 1 \,|\, A = 1) = 0.25$ inducing the agent to take the suboptimal action $a_F = 1$. Despite the forecast being observationally correct, it is not incentive-compatible with $U$.

However, if we use the utility score then every forecast with $F(Y = 1 \,|\, A = 0) > F(Y = 1 \,|\, A = 1)$ induces the optimal action $A = a^* = 0$ so it obtains the maximum expected score: see Figure 8a. (To improve visibility we have not depicted the global optima.) Since all forecasts that induce the action $A = 0$ obtain the same score, it is clear that the utility score is proper but not strictly proper. If we add any $\Delta < 0.25$ times the Brier score to the utility score then the resulting score becomes observationally strictly proper: the score is uniquely maximised for the correct value of $F(Y \,|\, A = 0)$; see Figure 8b.

## F.3 Scoring with divergence and IPW score

**Example 10.** Consider the setting of Example 7 where there is no positivity of $P_M(A \,|\, \text{do}(F))$. Recall that the Brier score is neither observationally proper nor counterfactually proper. However, if we consider the performative divergence related to the Brier score, as depicted in Figure 9a, we see that the correct forecast is an optimum, so it is proper. Moreover, for any forecast that attains the optimum we see that the action $A = 0$ is chosen (since the score does not depend on forecasts for $A = 1$) and every optimal forecast is correct for $A = 0$.

In Figure 9b we see that the IPW score is neither observationally nor counterfactually proper, similar as in Figure 7a.

When adding uniform exploration with probability $1/3$ we have positivity as in Example 8, we see that the the divergence and IPW score as depicted in Figure 9c and 9d are strictly proper: the correct forecast is the unique optimiser.

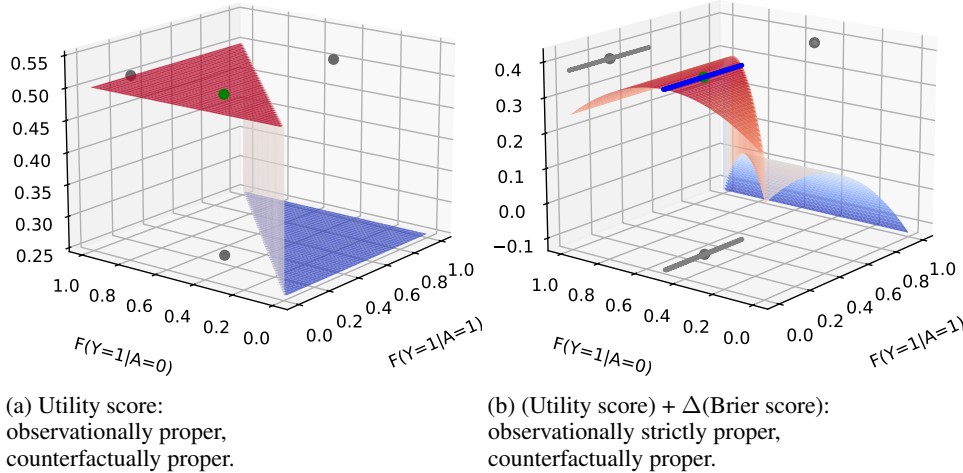

(a) Utility score: observationally proper, counterfactually proper.

(b) (Utility score) + $\Delta$(Brier score): observationally strictly proper, counterfactually proper.

Figure 8: Expected scores of Example 9. Both decision scoring rules are incentive compatible with utility $U$, since every forecast which attains maximum score induces the optimal action $A = a^* = 0$.

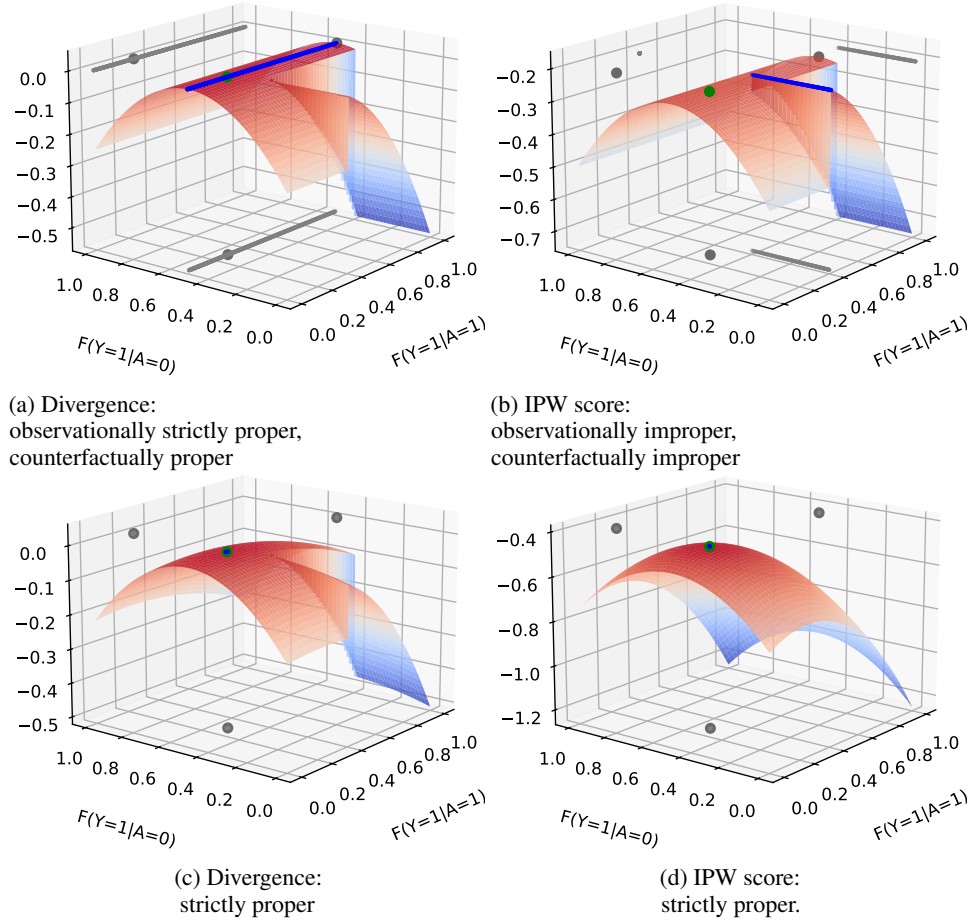

(a) Divergence: observationally strictly proper, counterfactually proper

(b) IPW score: observationally improper, counterfactually improper

(c) Divergence: strictly proper

(d) IPW score: strictly proper.

Figure 9: Plots related to Example 10. If there is no positivity (9a and 9b) then the divergence is observationally strictly proper and the IPW score is not observationally proper. If there is positivity (9c and 9d) then the divergence and IPW score are strictly proper.

## G   Bias and consistency of estimates of divergence and IPW

For general scoring rules $S$, plugin estimators for the divergence and IPW score are given by

$$\hat{D}_{\text{pc}}(F, M) = \frac{1}{n} \sum_{i=1}^{n} \left( S\left( \hat{P}(Y \mid A = a_i, \text{do}(F)), y_i \right) - S\left( F(Y \mid A = a_i), y_i \right) \right)$$

$$\hat{S}_{\text{IPW}}(F, M) = -\frac{1}{n} \sum_{i=1}^{n} \frac{S(F(Y \mid A = a_i), y_i)}{\hat{P}(A = a_i \mid \text{do}(F))},$$

using estimates of $\hat{P}(Y \mid A = a_i, \text{do}(F))$ and $\hat{P}(A = a_i \mid \text{do}(F))$. To investigate the variance and the reliability of these scoring methods at various sample sizes, we consider the mechanism of Example 8, so where $P_M(A \mid \text{do}(F))$ has full support for every $F \in \mathcal{P}_{\mathcal{Y}}$. In this setting, the unbiased estimator for the divergence is given by

$$\hat{D}_{\text{pc}}^{Brier} := \frac{1}{n} \sum_{i=1}^{n} (\hat{p}_{a_i} - f_{a_i})^2 - \frac{\hat{p}_{a_i}(1 - \hat{p}_{a_i})}{n_{a_i} - 1}, \tag{5}$$

where we write $n_a := \sum_{i=1}^{n} \mathbb{1}\{a_i = a\}$, $\hat{p}_a := \frac{1}{n_a} \sum_{i=1}^{n} y_i \mathbb{1}\{a_i = a\}$ and $f_a := F(Y = 1 \mid A = a)$.

We consider two forecasts: the correct forecast $F^*(Y \mid A = 0) = 0.5, F^*(Y \mid A = 1) = 0.25$, and an incorrect forecast $\tilde{F}(Y \mid A = 0) = 0.7, \tilde{F}(Y \mid A = 1) = 0.45$. For both methods, the correct forecast should obtain a lower score than the incorrect forecast. For the sample sizes $n \in \{2, 3, 4, 5, 6, 7, 8, 9, 10, 21, 46\}$ we simulate for both the correct and the incorrect forecast, 10.000 datasets of $(A_1, Y_1), ..., (A_n, Y_n) \sim P_M(A, Y \mid \text{do}(F))$, for $F = F^*$ and $F = \tilde{F}$. We subsequently estimate on every dataset the divergence and IPW score, and from those 400 repetitions we take the median and 5 and 95-percentiles. The results are depicted in Figure 10. This verifies that the plugin estimators for thr Brier divergence and IPW Brier score are biased but asymptotically unbiased, and that the estimator (5) for the Brier score is indeed unbiased.

These experiments can be run within 15 seconds on an Apple M2 Pro processor with 16GB RAM. Python code to run these experiments is provided in supplementary files.

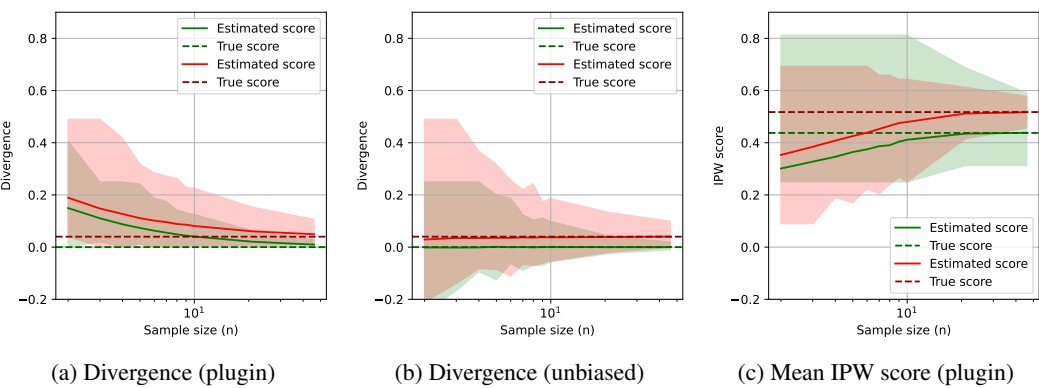

(a) Divergence (plugin)          (b) Divergence (unbiased)          (c) Mean IPW score (plugin)

Figure 10: Estimated divergence/IPW score (solid line) and true divergence/IPW score (dashed line), for a correct forecast (green) and an incorrect forecast (red), with 90% confidence intervals.

