# OpenReview forum: "Conditional Forecasts and Proper Scoring Rules for Reliable and Accurate Performative Predictions"
_NeurIPS.cc/2025/Conference — NeurIPS 2025 poster_

### Official Review · Reviewer_Tz7S · 2025-06-27

**Clarity:** 3
**Significance:** 3
**Originality:** 3
**Rating:** 5
**Confidence:** 3

**Summary:**

This paper studies the problem of how to formalize performative prediction and various issues and solutions that are particular to it (as opposed to other forms of forecasting). This problem is essentially that when a forecaster makes a forecast $F$, it alters the distribution of the outcomes. This is a problem because in such cases a forecast might never be correct in the sense that once it is predicted, it doesn't really match the "real-world" distribution of the event.

To avoid this issue, the authors formalize this problem via a causal mechanism using conditional forecasts and causal inference. Concretely, they focus on forecasts which are conditional on given actions and pose a causal graph on the problem where there is a path from $F$ to actions $A$ and then $Y$, and optionally one path from $F$ to $A$ directly.

Once this causal formalization has been introduced, the authors proceed to formalize various notions that have to do with these types of forecasts, such as three types of correctness (observational correctness, counterfactual correctness, and the combination of these two) and the notion of forecast invariance, which means that in the causal model $P(Y \mid A, \text{do}(F)) = P^M(Y\mid A)$. Forecast invariance is important as it solves to some degree the issue, which is a priori intuitive but also given the examples the authors provide. The authors also show that this notion is equivalent to d-separation of $Y$ and $F$ by $A$ in Theorem 1.

After completing the general setup, the authors proceed to discuss various problems in this setting.

The first one is the problem of proper scoring rules. The authors first straightforwardly extend the definitions for a reasonable proper scoring rule to the conditional setting and then show in Theorem 2 that all non-trivial proper scoring rules designed for the non-conditional setting fail for performative prediction. The intuition is that the forecaster may produce conditional forecasts that bias the actions to a world where it can more easily predict $Y$. Their first proposed solution is to align decisions by using utilities to create proper scoring rules. In Theorem 3, they show this score aligns with the goal of maximizing utility and is observationally proper in the $Y \mid A$ invariant setting. Theorem 4 then deals with some technicalities to make it strictly proper.

The second solution that they propose is batch scoring, which is essentially a divergence that measures the gap between the proper score of $P(Y \mid A=a, \text{do}(F))$ and the proper score of $F(Y\mid A= a)$, assuming the integral is with respect to $P(Y, A \mid \text{do}(F))$. They show that this scoring rule is proper, although it has the disadvantage of requiring access to $P(Y \mid A=a, \text{do}(F))$, which makes it impractical for many scenarios.

Finally, in Section 6, they propose an iterative procedure that extends these ideas to parameter estimation. In Theorem 6, they guarantee that this procedure is stable (so that it attains a fixed point) and is optimal in the sense of minimizing risk.

**Questions:**

I have no major questions beyond those stated in the weakness section.

**Ethical Concerns:**

["NO or VERY MINOR ethics concerns only"]

**Final Justification:**

The rebuttal did not change my view on the paper and all of my previous comments and positive opinion  of the paper stands so I retain my acceptance score of 5.

**Limitations:**

yes

**Quality:**

4

**Strengths And Weaknesses:**

**Strengths**: I thought the paper was very clear and well-written. I really enjoyed reading it.
I think the contributions that the paper makes are interesting. To state some:
- The rigorous causal framing to ground the framework is interesting and a good contribution as it clarifies the problem. While intuitive, formalizing is important.
- The impossibility result for classical scoring clearly articulates what might go wrong, and the authors do a good job explaining it to the reader both with examples and formally.
- The authors do a good job of both formalizing the criteria for proper scoring rules to work in this setting and also propose two ways of constructing proper scoring rules.
- The authors propose a method for parameter estimation in this setting and show that it reaches optimality and is well-behaved.

Weaknesses
In my opinion, there are no major weaknesses with the paper; however, some small ones are the following:

- Although the paper is well-written, I think there are some things that could be improved notationally. For example, the paper tends to use integrals all the time, which could be more cleanly expressed as expectations. This is especially the case in Section 6, where parsing the integral is quite dense and makes it hard to see what's happening at a glance. In this section as well, I felt that an algorithm box could have clarified things, although I understand the authors might have chosen not to do that due to space constraints.
- In Section 6, I didn't really understand the connection to score matching, and I think this could be spelled out more.
- One limitation of the batch scoring metric they propose is that it is hard to estimate from data in practical settings. I think it could be good if the authors devoted some more time to explaining this.
- I think one possible weakness of the paper is that a lot of the framework is fairly intuitive, and some of the contributions are mostly about formalizing things: coming up with the right definition and such. However, I think this work always needs to be done, so I wouldn't hold it strongly against the paper even if aspects of it are somewhat intuitive.
- A minor weakness is that this paper focuses on a very specific setting, so it is unclear to me how high the impact would be.

**Overall**
As I said, I really enjoyed reading the paper. While I think that due to being a bit dense, there could be more effort put into clarifying some notation and concepts, especially at the end, I think this is an interesting contribution.

---

> ### Author Rebuttal · Authors · 2025-07-30
>
> Dear reviewer, thank you for your helpful remarks and questions.
>
> **W1:**
>
> For example, before Definition 5, in the integral $\int S(F, y)P_M(d y | do(F))$ we want to explicitly denote the model $M$, the dependence $do(F)$, and the random variable $Y$ that we integrate over. Alternative notation would be $E_{P_M(Y | do(F))}\left[S(F, Y)\right]$, but since this does not seem much more intelligible we chose for the integral notation.
> We would like to spend part of the additional page to add an Algorithm box to Section 6.
>
> **W2:**
>
> Here, score matching is a name of the parameter estimation method using the Hyvärinen score $S(F_\theta, y) = \frac{\Delta \sqrt{f_\theta(y)}}{\sqrt{f_\theta(y)}}$ (where $f_\theta(y)$ denotes the density of the distribution $F_\theta(Y)$ and $\Delta$ denotes the Laplacian), as introduced by [1] (see also [2]). Perhaps the confusion stems from ‘propensity score matching’ in causal inference? This is an entirely different method. If deemed necessary, we can add a footnote to prevent confusion.
>
>
> **W3:**
>
> As also remarked in our rebuttal to Q1 of Reviewer z9DJ, we will add the important result that for the estimated divergence to be proper, one does not have to estimate it accurately; the only requirement is that an *unbiased* estimator is utilised. In that case, for any sample size that will be used to estimate the divergence, the forecaster's expected estimated score (divergence) equals the true expected score (divergence), and hence the method is proper.
>
> **W5:**
>
> Note that a setting where the forecast depends on covariates is analysed in the supplements, which greatly improves the applicability. We will move this section from the supplements to the appendix for greater visibility.
>
> Your final comment is similar to ‘weakness 1’ of Reviewer soxu. We would like to spend part of the additional page to add more elaborate motivations, explanations and examples to the paper.
>
> If any aspects remain unclear, we’d be happy to further elaborate during the discussion phase.
>
> [1] Hyvärinen, A. (2005). Estimation of Non-Normalized Statistical Models by Score Matching.
>
> [2] Dawid, A. P. and Musio, M. (2014). Theory and applications of proper scoring rules.

---

> > ### Comment · Reviewer_Tz7S · 2025-08-02
> >
> > I thank the authors for their response. I read the comments by the other reviewers and the responses and I think all of the responses that were provided make sense. I will keep my score because the weaknesses I mentioned were mostly small so although the authors did a good job of addressing them they don't change my overall opinion on the paper. I think the paper is a good contribution based on its formalization of various aspects of performative predictions. I don't give it a higher score as I think the paper would probably need to be more novel and focus less on formalization. However, I still believe it should be accepted as it was very enjoyable to read, it did a good job of explaining the setting and contributions and attained its goal of formalizing performative predictions.

---

### Official Review · Reviewer_soxu · 2025-06-27

**Clarity:** 3
**Significance:** 3
**Originality:** 3
**Rating:** 5
**Confidence:** 3

**Summary:**

The paper studies forecasting through the lens of performativity: it identifies limits of classical forecasting under performative feedback and proposes new tools for more reliable forecasting under performativity. It shows that classical proper scoring rules fail to be correct under performativity. Under suitable conditions, it also shows the existence of proper incentive-compatible scoring rules. Finally, the paper shows how correct forecasts can achieve performative stability and performative optimality simultaneously.

**Questions:**

1. Can you explain counterfactual correctness? I don't understand how we are able to talk about the conditional distribution Y|A=a, do(F) when P(A=a|do(F)) = 0. I don't understand this criterion conceptually or mathematically.
2. Is Theorem 1 a classical result?
3. In Theorem 6, \theta_{t+1} is performatively stable and optimal for *all* t, correct?

**Ethical Concerns:**

["NO or VERY MINOR ethics concerns only"]

**Final Justification:**

I was positive about this paper from the beginning. I'm arguably even more positive now after the discussion phase, knowing that Theorem 1 is novel. The paper is original and well-motivated.

**Limitations:**

Yes, these are explicitly discussed in Section 7.

**Paper Formatting Concerns:**

I would reduce the amount of text in footnotes. On page 4 especially, it is quite overwhelming.

**Quality:**

3

**Strengths And Weaknesses:**

This work is very well-motivated. In a lot of problems motivating performative prediction the goal is to make accurate forecasts; for example, in policy problems, such as when we are predicting a distribution of future covid cases. The paper establishes a clear foundation for studying forecasting under performativity, and it establishes basic first results contrasting classical forecasting with forecasting under performative feedback.

The current results are quite abstract and not very practical and prescriptive, but I think they are a good starting point. The examples in the paper were very helpful to me for grounding the discussion, and I would appreciate it if there were even more such concrete examples where all variables and desirable criteria would be fully explained and instantiated.

It's worth mentioning some classical results on the possibility of accurate forecasts under performativity; for example, "Bandwagon and Underdog Effects and the Possibility of Election Predictions" by Herbert Simon (1965).

---

> ### Author Rebuttal · Authors · 2025-07-30
>
> Dear reviewer, thank you for your helpful remarks and questions.
>
> Reviewer Tz7S made a similar comment, that the paper is a bit dense. Due to space constraints we had to put visual examples in the supplements. We would like to spend part of the additional page to add more elaborate motivations, explanations and examples to the paper.
>
> Thank you for pointing us to the paper by Herbert Simon. We will add a reference (also to Grunberg, E. and Modigliani, F. (1954). The Predictability of Social Events.) to the paper.
>
> **Q1:**
>
> Conceptually, counterfactual correctness means that the conditional forecast is correct for all actions $a\in \mathcal{A}$ which *cannot* be observed. This should be contrasted with observational correctness: that the forecast is correct for all actions which *can* be observed. Mathematically, the causal mechanism prescribes a distribution $P_M(Y|do(A=a, F))$ for all possible values of $a \in \mathcal{A}$, so also if $P_M(A=a | do(F))=0$. In the setting that we consider there is no confounding between $A$ and $Y$, so we can pick a version of the conditional distribution which satisfies $P_M(Y | A=a, do(F)) = P_M(Y|do(A=a, F))$ for all $a\in\mathcal{A}$.
> We will elaborate on this nuance in the paper.
>
> **Q2:**
>
> To our knowledge, Theorem 1 is entirely novel.
>
> **Q3:**
>
> Well, it might be the case that for $t=1$ the parameter is incorrect and not immediately performatively stable, but if the next parameters $t>=2$ are obtained by repeated minimization of the divergence, then $\theta_t$ is stable and optimal for all $t>=2$.
>
> If any aspects remain unclear, we’d be happy to further elaborate during the discussion phase.

---

> > ### Comment · Reviewer_soxu · 2025-08-05
> >
> > Thanks for the response. It's cool to hear that Theorem 1 is novel. Seems like a very fundamental result!

---

### Official Review · Reviewer_DFQj · 2025-06-29

**Clarity:** 3
**Significance:** 3
**Originality:** 3
**Rating:** 5
**Confidence:** 3

**Summary:**

This paper addresses the problem of performative prediction, where forecasts influence the outcomes they aim to predict, undermining classical notions of correctness and proper scoring. The authors show that conditioning forecasts on suitable covariates restores well-posedness but still renders classical scoring rules inadequate. They prove an impossibility result for standard scoring and propose alternative approaches—decision-theoretic and batch-scoring methods—that enable truthful and stable forecasting in these settings. The framework also resolves instability in parameter estimation identified in prior work.

**Questions:**

1. On page 8, paragraph below Algorithm 1. The authors mention that Chen and Kash assume that P_M(A) does not depend on the forecast. The distribution on the actions in Chen and Kash indeed depends on the forecast, according to my understanding. Their notation uses decision rules that take a forecast and output a distribution of actions. Hence, it is unclear how Theorem 5 generalises results in Chen and Kash. Perhaps the contribution of Theorem 5 needs to be clarified in the context of prior work.

2. Authors mention that to elicit correct forecasts, P(A|do(F)) needs to have full support, similar to previous results in Chen and Kash et al, which assumes that the random decision rule also has full support. How related is this assumption to randomised controlled trials?

**Ethical Concerns:**

["NO or VERY MINOR ethics concerns only"]

**Final Justification:**

I remain very positive about this paper. The rebuttal addressed my concerns. Thank you.

**Limitations:**

As mentioned above, limited empirical evaluation of the theoretical results proposed.

**Quality:**

3

**Strengths And Weaknesses:**

Strengths And Weaknesses:
- The authors propose an elegant formalisation of a hard problem, combining ideas from several fields of causality, scoring rules, and performitivity in a novel way.
- The main strength of the paper is its theoretical contributions on the characterisation of structure in which perfomitivity influences the real outcome and hence truthful scoring of the forecast.
- What is the connection between the formalisation proposed by the authors and the one proposed by Chen and Kash et al [1]? The way I see it, their notions of observationally strictly proper and counterfactually strictly proper and correct are related to the notion of quasi-strict properness and strict properness of forecasts in Chen and Kash et al. Placing the current formalisation in the context of prior work is important.
- Minimal evaluation of the theoretical results proposed by the authors. While the theoretical results are significant, it is still unclear how the proposed characterization of proper scoring rules is implemented in practice with respect to repeated risk minimization. Evaluation on the credit scoring dataset [3] similar to Perdomo et al [3] would strengthen the contributions of the paper even further.

1. Chen and Kash et al. Information Elicitation for Decision Making
2. Give me Some Credit. https://www.kaggle.com/c/GiveMeSomeCredit/data, 2012.
3. Perdomo et al Performative Prediction, ICML 2020

---

> ### Author Rebuttal · Authors · 2025-07-30
>
> Dear reviewer, thank you for your helpful remarks and questions.
>
> **S&W, point 3:**
>
> We agree that placing our formalisation in the context of prior work is important. To this end, we already make this comparison in footnote 9: “In a decision-theoretic context, [3] refer to observationally strictly proper and counterfactually proper scoring rules as `quasi-strictly proper' scoring rules.” (We think that separately considering observational and counterfactual properness provides a clearer language.) We will add a reference to [1] in the footnote.
>
> **Q1:**
>
> We don’t claim that Theorem 5 generalises [2]: it proves strict properness of the performative divergence, which is a different notion than the IPW score as considered by [2]. As far as we know, we are the first to consider the performative divergence, so Theorem 5 is entirely novel.\
> For completeness, we have proven in Theorem 9 in Appendix B that the IPW score (as considered by [2]) is strictly proper, and in the paper we claim that Theorem 9 generalises [2]. We realise that we have initially proven the same result as [2], albeit in a more explicit formalism.
> However, inspired by Q1 of Reviewer z9DJ we will add a corollary to Theorem 9 that unbiased estimates of the IPW score render it proper (see our rebuttal to Reviewer z9DJ), so this shows wider applicability of [2] in settings where $P(A|do(F))$ is not *known* to the principal but instead where the IPW score can be unbiasedly *estimated*, ultimately justifying the claim that we generalise [2].\
> We will better emphasize which results are novel and which results already exist, and we will adjust our claim of generalising [2] appropriately.
>
>
> **Q2:**
>
> If A is determined in an RCT then $F$ typically has no effect on $A$, so $P_M(A | do(F)) = P_M(A)$. In this case the forecast is not performative, and a classical strictly proper (marginal) scoring rule is observationally strictly proper for conditional forecasts (Appendix A, Theorem 7).\
> (If the reviewer is interested in a motivation of the assumption that $P_M(A | do(F))$ has full support, see also our rebuttal to Reviewer z9DJ, Q2.)
>
> If any aspects remain unclear, we’d be happy to further elaborate during the discussion phase.
>
>
> [1] Chen and Kash (2011). Information Elicitation for Decision Making.
>
> [2] Chen, Kash, Ruberry and Shnayder (2011). Decision Markets with Good Incentives.
>
> [3] Othman and Sandholm (2010). Decision Rules and Decision Markets.

---

> > ### Comment · Reviewer_DFQj · 2025-08-08
> >
> > Dear authors,
> >
> > Thank you for your response and clarification. In general, I remain positive about this paper.

---

### Official Review · Reviewer_z9DJ · 2025-06-30

**Clarity:** 3
**Significance:** 2
**Originality:** 3
**Rating:** 3
**Confidence:** 4

**Summary:**

this paper studies how to design scoring rules to elicit correct forecasts in the performative prediction setting. To overcome the impossibility described in self-defeating prophecies, the authors considers a causal framework where a conditional forecast F(Y|A) affects the conditioning variable A and the target variable Y, with a focus on the case where the conditional distribution Y|A is forecast-invariant --- which they show in Theorem 1 to be an equivalent condition for causal models to possess correct forecasts. The paper shows that even when Y|A is forecast invariant, classical proper scoring rules might no longer be proper in the performative setting. The authors then consider two approaches to construct proper/"incentive compatible" scoring rules, one is for decision-theoretic class of causal models, and is based on the combination of utility score and strictly proper scoring rule, under the assumption of a nonzero gap between optimal and suboptimal actions. Another is based on divergence (difference between entropy S(P,P) and score S(P,Y)), which avoids penalizing predictors that correctly predicts a high-entropy target, but only works in the batch setting where the evaluator has access to a large sample of events (A,Y) to estimate the entropy. Finally, the authors study parameter estimation, and showed that using divergence as the loss function results in performatively stable and optimal solutions when A|F has full support.

**Questions:**

1. Could you comment on the finite-sample guarantees of the divergence-based approach (Algorithm 1)? Please see weakness 3 -- did I miss anything?
2. Could you motivate the utility gap assumption in theorem 4 and the full support assumption in Theorem 6?

**Ethical Concerns:**

["NO or VERY MINOR ethics concerns only"]

**Final Justification:**

The authors' rebuttal partially addressed my concerns regarding the finite-sample properness of Thm 5 by proposing a sufficient condition --- the existence of an unbiased estimator for the entropy term. However, such unbiased estimators are not guaranteed to exist in general, and the authors only provided a few examples without a more general characterization. My concerns regarding the full support assumption also remains --- although being sharp, it is a quite strong assumption that leads to unnatural learning results. Although I appreciate the contributions of this paper, I think these two issues limit the applicability of the results in the batch setting and the parameter estimation setting.

**Limitations:**

yes

**Quality:**

2

**Strengths And Weaknesses:**

Strengths:

The paper studies the important question of how to evaluate and incentivize forecasts in a causal system where the target distribution is not fixed but is a response to the forecast itself. The authors identify conditional forecast as the correct notion that bypasses impossibility results, and developed some theory towards restoring proper scoring rules in this setting. The problem is theoretically interesting and practically relevant.

Weaknesses:
1. At a high level, while the paper covers multiple aspects of performative prediction (decision-theoretic, batch setting, parameter estimation), the theoretical depth for each aspect is limited. Most results are relatively straightforward extensions of classical scoring rule theory. The contributions feel more like a collection of observations rather than deep theoretical insights.
2. regarding the decision-theoretic approach: The existence of a utility gap might require some motivation.
3. regarding the divergence-based approach: The divergence used in Def 7 is guaranteed to be proper, but requires knowledge of the conditional distribution to compute the divergence, which might be impractical (especially when P(A|F) has full support, which implies that the principal already knows all the conditional distributions and would not need an external forecaster). While the finite-sample version in Algorithm 1 attempts to address this, the estimation of entropy introduces bias in the finite-sample regime, which might change the forecaster's incentives. For example, consider the Brier score and Y|A,do(F) ~ Bernoulli(p), the expected estimated entropy is (n-1)/n times the true entropy. This means that even though the performative divergence is proper in theory (Definition 7), the finite-sample approximation used in practice may not preserve properness.
4. regarding the parameter estimation results: Thm 6 requires A|Y to have full support, which may be unrealistic. This assumption (together with the fact that using divergence requires access to true distribution) also mildly trivialize the problem and makes it such that a single step of updating $\theta_t$ already yields the optimal solution, which seems far from practice.

---

> ### Author Rebuttal · Authors · 2025-07-30
>
> Dear reviewer, thank you for your helpful remarks and questions.
>
> **Q1:**
>
> Thank you for this interesting question. Indeed, if one utilises a biased estimator for the divergence, then the method need not be proper anymore. We will make a note of this in the main paper and provide examples in the supplements, similar to Figure 6 but then with biased estimates of the divergence and IPW score.
> However, due to your question we got the important insight that in some cases **this problem can be fully resolved: the estimated divergence is proper if one uses an unbiased estimator for it.** The forecaster optimizes the expected estimated divergence, which is equal to the true divergence in case the estimator is unbiased. This does not even require a large sample: any sample size for which the unbiased estimator is well-defined ($n>=2$ in your Bernoulli-Brier example) will do. In particular, the principal does not have to *know* the distribution $P(Y|A, do(F))$ – an unbiased estimate of the divergence is sufficient. We will:
> - remark that the method is not necessarily proper if one uses a biased estimator for the divergence;
> - add a corollary of Theorem 5 that the (unbiased) estimated divergence is proper for any sample size for which the estimator is well-defined;
> - make appropriate changes to Algorithm 1 to use an estimator of the divergence (and not the plugin estimator);
> - give an example of the unbiased estimator for the divergence based on the brier score;
> - note that the existence of an unbiased estimator is not guaranteed for certain choices of scoring rules (e.g. KL divergence)
> - add a corollary to Theorem 9 in Appendix B that the (unbiased) estimated IPW score is proper for any sample size (improving the applicability of the results of [1]);
> - change Figure 7 in the supplements to use an unbiased estimator for the divergence.
>
> **Q2:**
> - In Theorem 4, the principal may for example assume a utility gap if $A$ is binary and denotes taking an action which definitely has some effect on the utility, but it’s unclear whether it will be good or bad (like rolling out a bold marketing campaign, performing a risky operation, etc). The principal might assume that the expected utility under that action will surely differ from expected utility when not taking that action, it’s just unclear whether the utility would improve or not. In such a setting, the principal might want to elicit a forecast of what the effect of the action will be, in which case Theorem 4 gives a scoring rule to make that forecast truthful. We will add this motivation of the utility gap to the paper.
> - In Theorem 6, the assumption of having full support can be for example satisfied if actions $A$ are taken based on more information than the forecast only (or the features that the forecast take into account, as considered in the Supplements, section 5). For example, a doctor can decide to operate based on conversations with the patient -- information which is not available to the forecaster.\
> We note that the assumption of having full support is even *necessary* for having performative stability: if $\mathcal{A} = \mathbb{N}$, one could otherwise construct a model $M$ such that for every $\theta_t$ obtained trough retraining we have $\mathrm{supp}[P_M(A | do(F_{\theta_t}))] = ${$1, …, t+1$}  (i.e. the support of A increases for every newly obtained parameter) and $\theta_{t+1} \neq \theta_t$, so we don’t have performative stability. We will prove the necessity of the assumption of full support in the appendix.\
> Although unapplicable in some scenario’s, we regard the setting of this theorem as a nontrivial but fairly general model class in which performativity can be completely mitigated. There might be other regularity assumptions on the model class (e.g. of the map $F\mapsto P(A, Y | do(F))$, as considered by [2]) which do not require the assumptions of full support or forecast-invariance to obtain (approximate) performative optimality or stability. We leave this to future work, and will make a note of this in the discussion section.
>
>
> We hope that these further explanations and insights resolve some of your issues with the paper. If any aspects remain unclear, we’d be happy to further elaborate during the discussion phase.
>
> [1] Chen, Kash, Ruberry, and Shnayder. (2011). Decision Markets with Good Incentives.
>
> [2] Perdomo, Zrnic, Mendler-Dünner, and Hardt. (2020). Performative Prediction.

---

> > ### Comment · Reviewer_z9DJ · 2025-08-03
> >
> > Thank you for the detailed response.
> >
> > For Q1, I appreciate the discussion on unbiased estimators and it partially addressed my concerns regarding the properness of Algorithm 1. But I remain somewhat unconvinced, as the authors noted that unbiased estimators of the entropy do not always exist. So it's unclear to what extent Thm 5 extends to the finite-sample setting.
> >
> > For Q2, I'm not fully convinced by the argument that full support is necessary for the existence of a performatively stable parameter. My understanding is that necessity would require that for *every* model without full support, there is no performatively stable point; whereas your argument only constructs a single example. I understand that one has to make some assumptions to rule out such examples, but this alone does not justify why full support is the only reasonable assumption to make, especially since full support is kind of a strong assumption.

---

> > > ### Author Response · Authors · 2025-08-04
> > >
> > > Thank you very much for your further comments.
> > >
> > > **Q1:**
> > >
> > > We don’t fully characterise when Theorem 5 extends to the finite-sample setting, but we give a sufficient condition: the existence of an unbiased estimator for the divergence (or IPW score). This is not so restrictive: it exists in your Bernoulli-Brier example, but for example also for continuous $Y$ with the CRPS score (or Energy score for multivariate $Y$; we will mention this in the paper). It leaves open questions whether for other sample space - scoring rule combinations there exist proper scoring methods, but we hope that our rather widely applicable solution to this problem is deemed of sufficient interest to the reviewer.
> > >
> > > **Q2:**
> > >
> > > If besides $Y|A$ being forecast invariant you place no further restrictions on the model class, then the assumption of full support is necessary to ensure that *for all models satisfying your assumptions*, retraining is performatively stable. We should have mentioned the “for all models” quantifier in our rebuttal. Perhaps it would be better to refer to this assumption as *sharp*: without this assumption the theorem wouldn’t hold.\
> > > In our rebuttal we note that **we don’t claim that full support is the only reasonable assumption to make**, but we leave the investigation of other sufficient conditions for (approximate) performative stability and optimality to future work.
> > >
> > > We’d be happy to provide further elaboration, if necessary. Thank you very much for your thoughtful engagement!

---

> > > > ### Comment · Reviewer_z9DJ · 2025-08-05
> > > >
> > > > Thank you for the clarification. I appreciate the discussion, but remain unconvinced about the scope of Thm 5 (since the authors only provide a few examples where the unbiasedness holds, without a general characterization). And I also find the full support assumption quite strong --- it implies that stability and optimality can be achieved in a single step, which is unlikely to hold in practice. Improving both would make the results more applicable.

---

### Decision · Program_Chairs · 2025-09-17

**Decision:**

Accept (poster)

**Comment:**

This paper explores truthful elicitation and proper scoring rules in the context of performativity, where a forecast can influence the outcome it predicts. It demonstrates that traditional scoring rules are ineffective in this setting. The paper presents a general impossibility result and proposes two solutions: (1) In decision-theoretic settings, truthful and optimal scoring is achievable if forecasts are separating. (2) Batch-scoring methods, which use empirical divergence or inverse probability weighting, can produce accurate forecasts. These findings are then applied to parameter estimation and conditional forecasts, showing that proper scoring rules can lead to performatively stable estimation of correct parameters.

The reviews are overall positive. Nevertheless, Reviewer `z9DJ` remains skeptical about the finite-sample properness of Theorem 5 and the full support assumption. They cited that "*these two issues limit the applicability of the results in the batch setting and the parameter estimation setting.*" I find the concerns raised by Reviewer z9DJ reasonable. On the other hand, Reviewers `DFQj`, `soxu`, and `Tz7S` are in favour of accepting this paper, praising its originality and novelty. However, Reviewer `Tz7S` also pointed out that a lot of effort went into 'formalising things' rather than addressing non-trivial challenges of the problem. Regardless, Reviewer `soxu` said, "*they are a good starting point*" and appreciated the novelty of Theorem 1, which is corroborated by Reviewer `Tz7S` who stated, "*this work [formalising things] always needs to be done, so I wouldn't hold it strongly against the paper even if aspects of it are somewhat intuitive.*" This last comment seems to align with the concern raised by Reviewer `z9DJ`.

After reading the paper, it's clear that, despite some limitations, the paper has made a significant contribution in truthful elicitation of performative forecasts, elucidating the important challenges from the novel causal perspective and providing preliminary solutions to mitigate some of these challenges. This paper is arguably one of the few papers to thoroughly study the relationship between probabilistic forecasts, proper scoring rules, and performativity from the causal perspective. The main contribution is the new theoretical framework, while other areas, like the applicability of the results to batch processing and parameter estimation, remain less developed. As a result, it deserves a presentation at the conference.